# Unpacking Information Bottlenecks: Surrogate Objectives for Deep Learning

## Abstract

The Information Bottleneck principle offers both a mechanism to explain how deep neural networks train and generalize, as well as a regularized objective with which to train models. However, multiple competing objectives are proposed in the literature, and the information-theoretic quantities used in these objectives are difficult to compute for large deep neural networks, which in turn limits their use as a training objective. In this work, we review these quantities, and compare and unify previously proposed objectives, which allows us to develop surrogate objectives more friendly to optimization without relying on cumbersome tools such as density estimation. We find that these surrogate objectives allow us to apply the information bottleneck to modern neural network architectures. We demonstrate our insights on MNIST, CIFAR-10 and Imagenette with modern DNN architectures (ResNets).

## 1 Introduction

The Information Bottleneck (IB) principle, introduced by Tishby et al. (2000), proposes that training and generalization in deep neural networks (DNNs) can be explained by information-theoretic principles (Tishby and Zaslavsky, 2015; Shwartz-Ziv and Tishby, 2017; Achille and Soatto, 2018a). This is attractive as the success of DNNs remains largely unexplained by tools from computational learning theory (Zhang et al., 2016; Bengio et al., 2009). The IB principle suggests that learning consists of two competing objectives: maximizing the mutual information between the latent representation and the label to promote accuracy, while at the same time minimizing the mutual information between the latent representation and the input to promote generalization. Following this principle, many variations of IB objectives have been proposed (Alemi et al., 2016; Strouse and Schwab, 2017; Fischer and Alemi, 2020; Fischer, 2020; Fisher, 2019; Gondek and Hofmann, 2003; Achille and Soatto, 2018a), which, in supervised learning, have been demonstrated to benefit robustness to adversarial attacks (Alemi et al., 2016; Fisher, 2019) and generalization and regularization against overfitting to random labels (Fisher, 2019).

Whether the benefits of training with IB *objectives* are due to the IB *principle*, or some other unrelated mechanism, remains unclear (Saxe et al., 2019; Amjad and Geiger, 2019; Tschannen et al., 2019), suggesting that although recent work has also tied the principle to successful results in both unsupervised and self-supervised learning (Oord et al., 2018; Belghazi et al., 2018; Zhang et al., 2018; Burgess et al., 2018, among others), our understanding of how IB objectives affect representation learning remains unclear.

Critical to studying this question is the computation of the information-theoretic quantities[1] used. While progress has been made in developing mutual information estimators for DNNs (Poole et al., 2019; Belghazi et al., 2018; Noshad et al., 2019; McAllester and Stratos, 2018; Kraskov et al., 2004), current methods still face many limitations when concerned with high-dimensional random variables (McAllester and Stratos, 2018) and rely on complex estimators or generative models. This presents a challenge to training with IB objectives.

In this paper, we analyze information quantities and relate them to surrogate objectives for the IB principle which are more friendly to optimization, showing that complex or intractable IB objectives can be replaced with simple, easy-to-compute surrogates that produce similar performance and similar

---

[1]We shorten these to *information quantities* from now on.

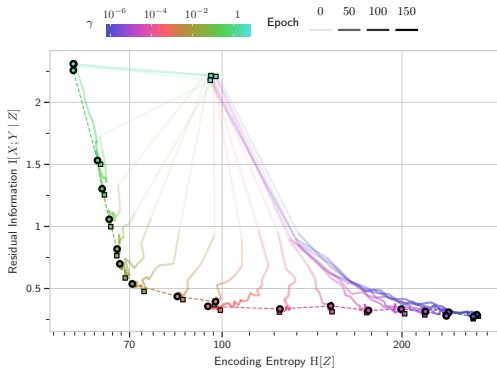

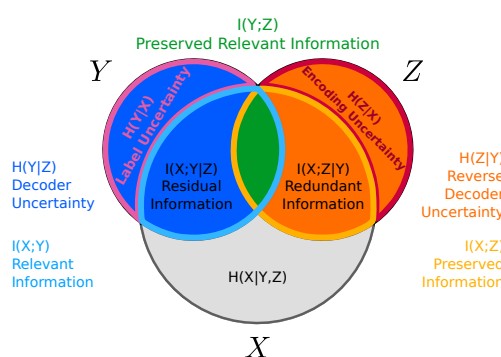

Figure 1: *Information plane plot of the training trajectories of ResNet18 models with our surrogate objective* $\min_\theta H_\theta[Y \mid Z] + \gamma \mathbb{E} \|Z\|^2$ *on Imagenette. Color shows $\gamma$; transparency the training epoch. Compression (Encoding Entropy ↓) trades-off with test performance (Residual Information ↓). See section 4.*

Figure 2: *Mickey Mouse I-diagram for the information quantities in the model* $p(x,y,z) = \hat{p}(x,y)\, p_\theta(z \mid x)$. X is for the data, Y for the labels, Z for the latent encodings. See section 2 for more details and section B.1 for a description of all the quantities. $I[Y; Z] = I[X; Y; Z]$ because $I[Y; Z \mid X] = 0$. Best viewed in color.

behaviour of information quantities over training. **Sections 2 & 3** review commonly-used information quantities for which we provide mathematically grounded intuition via information diagrams and unify different IB objectives by identifying two key information quantities, Decoder Uncertainty $H[Y \mid Z]$ and Reverse Decoder Uncertainty $H[Z \mid Y]$ which act as the main loss and regularization terms in our unified IB objective. In particular, **Section 3.2** demonstrates that using the Decoder Uncertainty as a training objective can minimize the training error, and shows how to estimate an upper bound on it efficiently for well-known DNN architectures. We expand on the findings of Alemi et al. (2016) in their variational IB approximation and demonstrate that this upper bound is equal to the commonly-used cross-entropy loss[2] under dropout regularization. **Section 3.3** examines pathologies of differential entropies that hinder optimization and proposes adding Gaussian noise to force differential entropies to become non-negative, which leads to new surrogate terms to optimize the Reverse Decoder Uncertainty. Altogether this leads to simple and tractable surrogate IB objectives such as the following, which uses dropout, adds Gaussian noise over the feature vectors $f(x; \eta)$, and uses an L2 penalty over the noisy feature vectors:

$$\min_\theta \mathbb{E}_{\substack{x,y \sim \hat{p}(x,y), \epsilon \sim \mathcal{N} \\ \eta \sim \text{dropout mask}}} \left[ -\log p(\hat{Y} = y \mid z = f_\theta(x; \eta) + \epsilon) + \gamma \|f_\theta(x; \eta) + \epsilon\|_2^2 \right]. \quad (1)$$

**Section 4** describes experiments that validate our insights qualitatively and quantitatively on MNIST, CIFAR-10 and Imagenette, and shows that with objectives like the one in equation (1) we obtain information plane plots (as in figure 1) similar to those predicted by Tishby and Zaslavsky (2015). Our simple surrogate objectives thus induce the desired behavior of IB objectives while scaling to large, high-dimensional datasets. We present **evaluations on CIFAR-10 and Imagenette images**[3].

Compared to existing work, we show that we can optimize IB objectives for well-known DNN architectures using standard optimizers, losses and simple regularizers, without needing complex estimators, generative models, or variational approximations. This will allow future research to make better use of IB objectives and study the IB principle more thoroughly.

## 2 BACKGROUND

**Information quantities & information diagrams.** We denote entropy $H[\cdot]$, joint entropy $H[\cdot, \cdot]$, conditional entropy $H[\cdot \mid \cdot]$, mutual information $I[\cdot; \cdot]$ and Shannon's information content $h(\cdot)$ (Cover and Thomas, 2012; MacKay, 2003; Shannon, 1948). We will further require the Kullback-Leibler divergence $D_{KL}(\cdot \| \cdot)$ and cross-entropy $H(\cdot \| \cdot)$. The definitions can be found in section A.1. We will use differential entropies interchangeably with entropies: equalities between them are preserved in the differential setting, and inequalities will be covered in section 3.3.

---

[2]This connection was assumed without proof by Achille and Soatto (2018a;b).

[3]Recently, Fischer and Alemi (2020) report results on CIFAR-10 and ImageNet, see section F.4.

Information diagrams (I-diagrams), like the one depicted in figure 2, clarify the relationship between information quantities: *similar to Venn diagrams, a quantity equals the sum of its parts in the diagram.* Importantly, they offer a grounded intuition as Yeung (1991) show that we can define a signed measure $\mu^*$ such that information quantities map to abstract sets and are consistent with set operations. We provide details on how to use I-diagrams and what to watch out for in section A.2.

**Probabilistic model.** We will focus on a supervised classification task that makes prediction $\hat{Y}$ given data X using a latent encoding Z, while the provided target is Y. We assume categorical Y and $\hat{Y}$, and continuous X. Our probabilistic model based on these assumptions is as follows:

$$p(x, y, z, \hat{y}) = \hat{p}(x, y)\, p_\theta(z \,|\, x)\, p_\theta(\hat{y} \,|\, z). \tag{2}$$

Thus, Z and Y are independent given X, and $\hat{Y}$ is independent of X and Y given Z. The data distribution $\hat{p}(x, y)$ is only available to us as an empirical sample distribution. $\theta$ are the parameters we would like to learn. $p_\theta(z \,|\, x)$ is the encoder from data X to latent Z, and $p_\theta(\hat{y} \,|\, z)$ the decoder from latent Z to prediction $\hat{Y}$. Together, $p_\theta(z \,|\, x)$ and $p_\theta(\hat{y} \,|\, z)$ form the discriminative model $p_\theta(\hat{y} \,|\, x)$:

$$p_\theta(\hat{y} \,|\, x) = \mathbb{E}_{p_\theta(z|x)}\, p_\theta(\hat{y} \,|\, z). \tag{3}$$

We can derive the cross-entropy loss $H(\hat{p}(y \,|\, x) \,\|\, p_\theta(\hat{Y} = y \,|\, x))$ (Solla et al., 1988; Hinton, 1990) by minimizing the Kullback-Leibler divergence between the empirical sample distribution $\hat{p}(x, y)$ and the parameterized distribution $p_\theta(x)\, p_\theta(\hat{y} \,|\, x)$, where we set $p_\theta(x) = \hat{p}(x)$. See section D.1.

**Mickey Mouse I-diagram.** The corresponding I-diagram for X, Y, and Z is depicted in figure 2. As some of the quantities have been labelled before, we try to follow conventions and come up with consistent names otherwise. *Section B.1 provides intuitions for these quantities, and section B.2 lists all definitions and equivalences explicitly.* For categorical Z, all the quantities in the diagram are positive, which allows us to read off inequalities from the diagram: only $I[X; Y; Z]$ could be negative, but as Y and Z are independent given X, we have $I[Y; Z|X] = 0$, and $I[X; Y; Z] = I[Y; Z] - I[Y; Z|X] = I[Y; Z] \geq 0$. Section 3.3 investigates how to preserve inequalities for continuous Z.

## 3 Surrogate IB & DIB objectives

### 3.1 IB Objectives

Tishby et al. (2000) introduce the IB objective as a relaxation of a constrained optimization problem: minimize the mutual information between the input *X* and its latent representation *Z* while still accurately predicting *Y* from *Z*. An analogous objective which yields deterministic *Z*, the Deterministic Information Bottleneck (DIB) was proposed by Strouse and Schwab (2017). Letting $\beta$ be a Lagrange multiplier, we arrive at the IB and DIB objectives:

$$\min I[X; Z] - \beta I[Y; Z] \quad \text{for IB, and} \quad \min H[Z] - \beta I[Y; Z] \text{ for DIB.} \tag{4}$$

This principle can be recast as a generalization of finding minimal sufficient statistics for the labels given the data (Shamir et al., 2010; Tishby and Zaslavsky, 2015; Fisher, 2019): it strives for minimality and sufficiency of the latent Z. Minimality is achieved by minimizing the Preserved Information $I[X; Z]$; while sufficiency is achieved by maximizing the Preserved Relevant Information $I[Y; Z]$. We defer an in-depth discussion of the IB principle to the appendix Section C.1. We discuss the several variants of IB objectives, and justify our focus on IB and DIB, in Section C.2.

The information quantities that appear in the IB objective are not tractable to compute for the representations learned by many function classes of interest, including neural networks; for example, Strouse and Schwab (2017) only obtain an analytical solution to their Deterministic Information Bottleneck (DIB) method for the tabular setting. Alemi et al. (2016) address this challenge by constructing a variational approximation of the IB objective, but their approach has not been applied to more complex datasets than MNIST variants. Belghazi et al. (2018) use a separate statistics network to approximate the mutual information, a computationally expensive strategy that does not easily lend itself to optimization.

In this section, we introduce and justify tractable surrogate losses that are easier to apply in common deep learning pipelines, and which can be scaled to large and high-dimensional datasets. We begin by proposing the following reformulation of IB and DIB objectives.

**Proposition 1.** *For IB, we obtain*

$$\arg\min I[X;Z] - \beta I[Y;Z] = \arg\min H[Y\,|\,Z] + \beta' \underbrace{I[X;Z\,|\,Y]}_{=H[Z|Y]-H[Z|X]}, \tag{5}$$

*and, for DIB,*

$$\arg\min H[Z] - \beta I[Y;Z] = \arg\min H[Y\,|\,Z] + \beta' H[Z\,|\,Y] = \arg\min H[Y\,|\,Z] + \beta'' H[Z] \tag{6}$$

*with* $\beta' := \frac{1}{\beta-1} \in [0,\infty)$ *and* $\beta'' := \frac{1}{\beta} \in [0,1)$. *The derivation can be found in section C.3.*

In the next sections, we show that Decoder Uncertainty $H[Y\,|\,Z]$ provides a loss term, which minimizes the training error, and DIB's Reverse Decoder Uncertainty $H[Z\,|\,Y]$ and IB's Redundant Information $I[X;Z\,|\,Y]$, respectively, provide a regularization term, which helps generalization. Another perspective can be found by relating the objectives to the Entropy Distance Metric introduced by MacKay (2003), which we detail in section C.4.

### 3.2 DECODER UNCERTAINTY $H[Y\,|\,Z]$

The Decoder Uncertainty $H[Y\,|\,Z]$ is the first term in our reformulated IB and DIB objectives, and captures the data fit component of the IB principle. This quantity is not easy to compute directly for arbitrary representations $Z$, so we turn our attention to two related entities instead, where we use $\theta$ as subscript to mark dependence on the model: the Prediction Cross-Entropy, denoted $H_\theta[Y\,|\,X]$ (more commonly known as the model's cross-entropy loss; see section D.1), and the Decoder Cross-Entropy, denoted $H_\theta[Y\,|\,Z]$. Noting that $h(x) = -\ln x$, we define these terms as follows:

$$H_\theta[Y\,|\,X] := H(\hat{p}(y\,|\,x) \,\|\, p_\theta(\hat{Y} = y\,|\,x)) = \mathbb{E}_{\hat{p}(x,y)} h\left(\mathbb{E}_{p_\theta(z|x)} p_\theta(\hat{Y} = y\,|\,z)\right) \tag{7}$$

$$H_\theta[Y\,|\,Z] := H(p(y\,|\,z) \,\|\, p_\theta(\hat{Y} = y\,|\,z)) = \mathbb{E}_{\hat{p}(x,y)} \mathbb{E}_{p_\theta(z|x)} h\left(p_\theta(\hat{Y} = y\,|\,z)\right). \tag{8}$$

Jensen's inequality yields $H_\theta[Y\,|\,X] \le H_\theta[Y\,|\,Z]$, with equality iff $Z$ is a deterministic function of $X$. The notational similarity[4] between $H_\theta[Y\,|\,Z]$ and $H[Y\,|\,Z]$ is deliberately suggestive: this cross-entropy bounds the conditional entropy $H[Y\,|\,Z]$, as characterized in the following proposition.

**Proposition 2.** *The Decoder Cross-Entropy provides an upper bound on the Decoder Uncertainty:*

$$H[Y\,|\,Z] \le H[Y\,|\,Z] + D_{KL}(p(y\,|\,z) \,\|\, p_\theta(\hat{y}\,|\,z)) = H_\theta[Y\,|\,Z], \tag{9}$$

*and further bounds the training error:*

$$p(\text{``}\hat{Y}\text{ is wrong''}) \le 1 - e^{-H_\theta[Y|Z]} = 1 - e^{-\left(H[Y|Z] + D_{KL}(p(y|z)\|p_\theta(\hat{y}|z))\right)}. \tag{10}$$

*Likewise, for* $H_\theta[Y\,|\,X]$ *and* $H[Y\,|\,X]$. *See section D.2 for a derivation.*

Hence, by bounding $D_{KL}(p(y\,|\,z) \,\|\, p_\theta(\hat{y}\,|\,z))$, we can obtain a bound for the training error in terms of $H[Y\,|\,Z]$. We examine one way of doing so by using optimal decoders $p_\theta(\hat{y}\,|\,z) := p(Y = \hat{y}\,|\,z)$ for the case of categorical $Z$ in section E.

Alemi et al. (2016) use the Decoder Cross-Entropy bound in equation (9) to variationally approximate $p(y\,|\,z)$. We make this explicit by applying the reparameterization trick to rewrite the latent $z$ as a parametric function of its input $x$ and some independent auxiliary random variable $\eta$, i.e. $f_\theta(x,\eta) \overset{D}{=} z \sim p_\theta(z\,|\,x)$, yielding

$$H[Y\,|\,Z] \le H_\theta[Y\,|\,Z] = \mathbb{E}_{\hat{p}(x,y)} \mathbb{E}_{p(\eta)} h\left(p_\theta(\hat{Y} = y\,|\,z = f_\theta(x;\eta))\right). \tag{11}$$

Equation (11) can be applied to many forms of stochastic regularization that turn deterministic models into stochastic ones, in particular dropout. This allows us to use modern DNN architectures as stochastic encoders.

**Dropout regularization** When we interpret $\eta$ as a sampled dropout mask for a DNN, DNNs that use dropout regularization (Srivastava et al., 2014), or variants like DropConnect (Wan et al., 2013a), fit the equation above as stochastic encoders. Monte-Carlo dropout (Gal and Ghahramani, 2016), for example, even specifically estimates the predictive mean $p_\theta(\hat{y}\,|\,x)$ from equation (3). The following result extends the observation by Burda et al. (2015) that sampling yields an unbiased estimator for the Decoder Cross-Entropy $H_\theta[Y\,|\,Z]$, while it only yields a biased estimator for the Prediction Cross-Entropy $H_\theta[Y\,|\,X]$ (which it upper-bounds).

---

[4]This notation is compatible with $\mathcal{V}$-Entropy introduced by Xu et al. (2020).

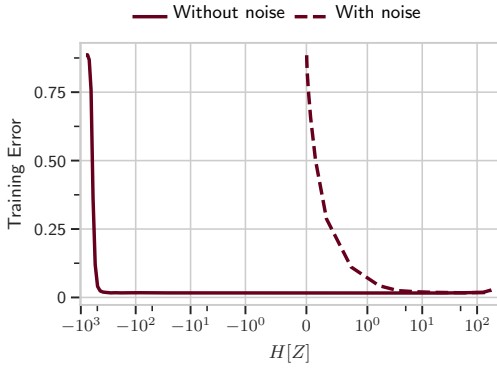
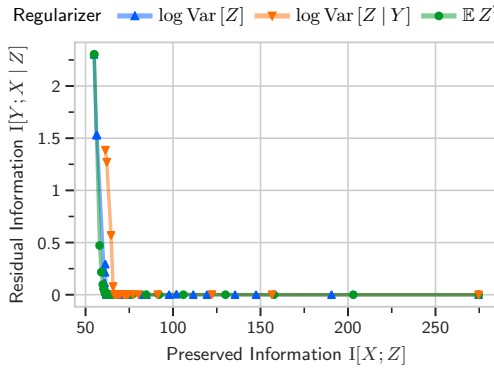

Figure 3: *Decreasing the entropy of a noise-free latent does not affect the training error.* (Though floating-point issues start affecting it negatively eventually.) When adding zero-entropy noise, the error rate increases as the entropy approaches zero. See section 3.3 and G.4 for more details.

Figure 4: *Information plane plot of the latent* Z *similar to* Tishby and Zaslavsky (*2015*) *but using a* ResNet18 *model on* CIFAR-10 *using the different regularizes from section 3.3.* A larger version can be found in figure G.1. See section 4 for more details. Best viewed in color.

**Corollary 1.** *Let* $x, y, z$ *and* $f_\theta$ *be defined as previously, with* $\eta$ *a sampled stochastic dropout mask. Then* $h\left(p_\theta(\hat{Y} = y \mid z = f_\theta(x; \eta))\right)$ *evaluated for a single sample* $\eta$ *is an unbiased estimator of the Decoder Cross-Entropy* $H_\theta[Y \mid Z]$*, and an estimator of an upper bound on the Prediction Cross-Entropy* $H_\theta[Y \mid X]$.

This distinction between the Decoder Cross-Entropy and the Prediction Cross-Entropy has been observed in passing in the literature, but not made explicit. Multi-sample approaches like Multi-Sample Dropout (Inoue, 2019), for example, optimize $H_\theta[Y \mid Z]$, while Importance Weighted Stochastic Gradient Descent (Noh et al., 2017) optimizes $H_\theta[Y \mid X]$. Dusenberry et al. (2020) observe empirically in the different context of rank-1 Bayesian Neural Networks that optimizing $H_\theta[Y \mid Z]$ instead of $H_\theta[Y \mid X]$ is both easier and also yields better generalization performance (NLL, accuracy, and ECE), while they also put forward an argument for why the stochastic gradients for $H_\theta[Y \mid Z]$ might benefit from lower variance. We empirically compare training with either cross-entropy in section G.3.3 and show results in figure G.11 in the appendix. We conclude this section by highlighting that $H[Y \mid Z]$ is therefore already minimized in modern DNN architectures that use dropout together with a cross-entropy loss. This means that, at least for one half of our reformulation of the IB objective, we can apply off-the-shelf, scalable objectives and optimizers for its minimization.

### 3.3 SURROGATES FOR THE REGULARIZATION TERMS

In the previous section, we have examined how to tractably estimate the error minimization term $H[Y \mid Z]$. In this section, we will examine tractable optimization of the regularization terms $H[Z \mid Y]$ and $I[X; Z \mid Y]$, respectively. We discuss how to minimize entropies meaningfully and show how this unifies DIB and IB via the inequality $I[X; Z \mid Y] \leq H[Z \mid Y] \leq H[Z]$ before providing tractable upper-bounds for $H[Z \mid Y]$ and $H[Z]$.

**Differential entropies** In most cases, the latent Z is a continuous random variable in many dimensions. Unlike entropies on discrete probability spaces, differential entropies defined on continuous spaces are not bounded from below. This means that the DIB objective is not guaranteed to have an optimal solution and allows for pathological optimization trajectories in which the variance of the latent $Z$ can be scaled to be arbitrarily small, achieving arbitrarily high-magnitude negative entropy. We provide a toy experiment demonstrating this in section G.4.

Intuitively, one can interpret this issue as being allowed to encode information in an arbitrarily-small real number using infinite precision, similar to arithmetic coding (MacKay, 2003; Shwartz-Ziv and Tishby, 2017)[5]. In practice, due to floating point constraints, optimizing DIB naively will invariably end in garbage predictions and underflow as activations approach zero. It is therefore not desirable

---

[5]Conversely, MacKay (2003) notes that without upper-bounding the "power" $\mathbb{E}_{p(z)} Z^2$, all information could be encoded in a single very large integer.

for training. This is why Strouse and Schwab (2017) only consider analytical solutions to DIB by evaluating a limit for the tabular case. MacKay (2003) proposes the introduction of noise to solve this issue in the application of continuous communication channels.

However, here we propose adding specific noise to the latent representation to lower-bound the conditional entropy of $Z$, which allows us to enforce non-negativity across all IB information quantities as in the discrete case and transport inequalities to the continuous case: for a continuous $\hat{Z} \in \mathbb{R}^k$ and independent noise $\epsilon$, we set $Z := \hat{Z} + \epsilon$; the differential entropy then satisfies $H[Z] = H[\hat{Z} + \epsilon] \geq H[\epsilon]$; and by using *zero-entropy noise* $\epsilon \sim \mathcal{N}(0, \frac{1}{2\pi e} I_k)$ specifically, we obtain $H[Z] \geq H[\epsilon] = 0$.

**Proposition 3.** *After adding zero-entropy noise, the inequality $I[X; Z \mid Y] \leq H[Z \mid Y] \leq H[Z]$ also holds for continuous Z, and we can minimize $I[X; Z \mid Y]$ in the IB objective by minimizing $H[Z \mid Y]$ or $H[Z]$, similarly to the DIB objective.*

Strictly speaking, zero-entropy noise is not necessary for optimizing the bounds: any Gaussian noise is sufficient, but zero-entropy noise is aesthetically appealing as it preserves inequalities from the discrete setting. In a sense, this propostion bounds the IB objective by the DIB objective. However, adding noise changes the optimal solutions: whereas DIB in Strouse and Schwab (2017) leads to hard clustering in the limit, adding noise leads to soft clustering when optimizing the DIB objective, as is the case with the IB objective. We show in section F.6 that minimizing the DIB objective with noise leads to soft clustering (for the case of an otherwise deterministic encoder). Altogether, in addition to Shwartz-Ziv and Tishby (2017), we argue that noise is essential to obtain meaningful differential entropies and to avoid other pathological cases as described further in section F.7.

It is not generally possible to compute $H[Z \mid Y]$ exactly for continuous latent representations $Z$, but we can derive an upper bound. The maximum-entropy distribution for a given covariance matrix $\Sigma$ is a Gaussian with the same covariance.

**Proposition 4.** *The Reverse Decoder Uncertainty can be approximately bounded using the empirical variance $\widehat{\mathrm{Var}}[Z_i \mid y]$:*

$$H[Z \mid Y] \leq \mathbb{E}_{\hat{p}(y)} \sum_i \frac{1}{2} \ln(2\pi e \, \mathrm{Var}[Z_i \mid y]) \approx \mathbb{E}_{\hat{p}(y)} \sum_i \frac{1}{2} \ln(2\pi e \, \widehat{\mathrm{Var}}[Z_i \mid y]), \tag{12}$$

*where $Z_i$ are the individual components of $Z$. $H[Z]$ can be bounded similarly. More generally, we can create an even looser upper bound by bounding the mean squared norm of the latent:*

$$\mathbb{E} \, \|Z\|^2 \leq C' \Rightarrow H[Z \mid Y] \leq H[Z] \leq C, \tag{13}$$

*with $C' := \frac{k e^{2C/k}}{2\pi e}$ for $Z \in \mathbb{R}^k$. See section F.2 for proof.*

**Surrogate objectives** These surrogate terms provide us with three different upper-bounds that we can use as surrogate regularizers. We refer to them as: conditional log-variance regularizer ($\log \mathrm{Var}[Z \mid Y]$), log-variance regularizer ($\log \mathrm{Var}[Z]$) and activation $L_2$ regularizer ($\mathbb{E} \, \|Z\|^2$). We can now propose the main results of this paper: IB surrogate objectives that reduce to an almost trivial implementation using the cross-entropy loss and one of the regularizers above while adding zero-entropy noise to the latent $Z$.

**Theorem 1.** *Let $Z$ be obtained by adding a single sample of zero-entropy noise to a single sample of the output z of the stochastic encoder. Then each of the following objectives is an estimator of an upper bound on the IB objective. In particular, for the surrogate objective $\mathbb{E} \, \|Z\|^2$, we obtain:*

$$\min H(p(y \mid z) \| p_\theta(\hat{Y} = y \mid z)) + \gamma \|z\|^2; \tag{14}$$

*for $\log \mathrm{Var}[Z \mid Y]$:*

$$\min H(p(y \mid z) \| p_\theta(\hat{Y} = y \mid z)) + \gamma \, \mathbb{E}_{\hat{p}(y)} \sum_i \frac{1}{2} \ln(2\pi e \, \widehat{\mathrm{Var}}[Z_i \mid y]); \tag{15}$$

*and for $\log \mathrm{Var}[Z]$:*

$$\min H(p(y \mid z) \| p_\theta(\hat{Y} = y \mid z)) + \gamma \sum_i \frac{1}{2} \ln(2\pi e \, \widehat{\mathrm{Var}}[Z_i]). \tag{16}$$

For the latter two surrogate regularizers, we can relate their coefficient $\gamma$ to $\beta'$, $\beta''$ and $\beta$ from section 3. However, as regularizing $\mathbb{E} \, \|Z\|^2$ does not approximate an entropy directly, its coefficient does not relate to the Lagrange multiplier of any fixed IB objective. We compare the performance of these objectives in section 4.

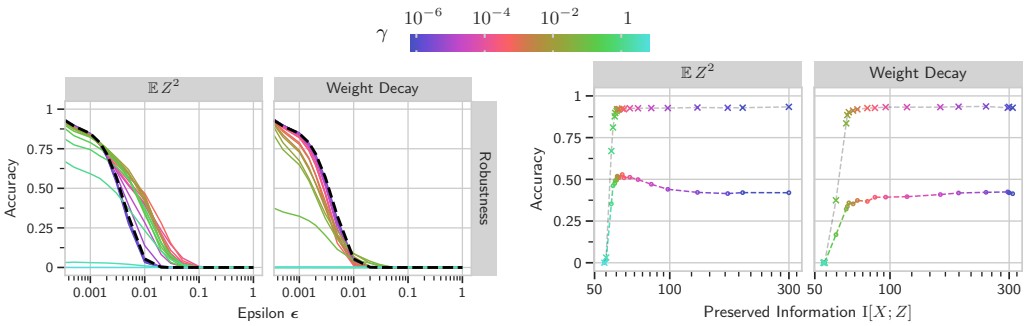

(a) *Robustness for different attack strengths $\epsilon$.* The dashed black line represents a model trained only with cross-entropy and no noise injection. We see that models trained with the surrogate IB objective (colored by $\gamma$) see improved robustness over a model trained only to minimize the cross-entropy training objective (shown in black) while the models regularized with weight-decay actually perform worse.

(b) *Average robustness over $\epsilon \in [0, 0.1]$ compared to normal accuracy for different amounts of Preserved Information.* ∘ markers show robustness. × markers show the normal accuracy. We see that robustness depends on the Preserved Information. If the latent is compressesed too much, robustness (and accuracy) are low. If the latent is not compressed enough, robustness and thus generalization suffer.

Figure 5: *Adversarial robustness of ResNet18 models trained on CIFAR-10 with surrogate objectives in comparison to regularization with L2 weight-decay as non-IB method.* The robustness is evaluated using FGSM, PGD, DeepFool and BasicIterative attacks of varying $\epsilon$ values.

## 4 EXPERIMENTS

We now provide empirical verification of the claims made in the previous sections. Our goal in this section is to highlight two main findings: first, that our surrogate objectives obtain similar behavior to what we expect of exact IB objectives with respect to their effect on robustness to adversarial examples. In particular, we show that our surrogate IB objectives improve adversarial robustness compared to models trained only on the cross-entropy loss, consistent with the findings of Alemi et al. (2016). Second, we show the effect of our surrogate objectives on information quantities during training by plotting information plane diagrams, demonstrating that models trained with our objectives trade off between $I[X; Z]$ and $I[Y; Z]$ as expected. We show this by recovering information plane plots similar to the ones in Tishby and Zaslavsky (2015) and qualitatively examine the optimization behavior of the networks through their training trajectories. We demonstrate the scalability of our surrogate objectives by applying our surrogate IB objectives to the CIFAR-10 and Imagenette datasets, high-dimensional image datasets.

For details about our experiment setup, DNN architectures, hyperparameters and additional insights, see section G. In particular, empirical quantification of our observations on the relationship between the Decoder Cross-Entropy loss and the Prediction Cross-Entropy are deferred to the appendix due to space limitations as well as the description of the toy experiment that shows that minimizing H[Z | Y] for continuous latent Z without adding noise does not constrain information meaningfully and that adding noise solves the issue as detailed in section 3.3.

**Robustness to adversarial attacks** Alemi et al. (2016) and Fischer and Alemi (2020) observe that their IB objectives lead to improved adversarial robustness over standard training objectives. We perform a similar evaluation to see whether our surrogate objectives also see improved robustness. We train a fully-connected residual network on CIFAR-10 for a range of regularization coefficients $\gamma$ using our $\mathbb{E} \|Z\|^2$ surrogate objective; we then compare against a similar regularization method that does not have an information-theoretic interpretation: L2 weight-decay. We inject zero-entropy noise in both cases. After training, we evaluate the models on adversarially perturbed images using the FGSM (Szegedy et al., 2013), PGD (Madry et al., 2018), BasicIterative (Kurakin et al., 2017) and DeepFool (Moosavi-Dezfooli et al., 2016) attacks for varying levels of the perturbation magnitude parameter $\epsilon$. We also compare to a simple unregularized cross-entropy baseline (black dashed line). To compute overall robustness, we use each attack in turn and only count a sample as robust if it defeats them all. As depicted in figure 5, we find that our surrogate objectives yield significantly more robust models while obtaining similar test accuracy on the unperturbed data whereas weight-decay regularization reduces robustness against adversarial attacks. Plots for the other two regularizers can be found in the appendix in figure G.13 and figure G.14.

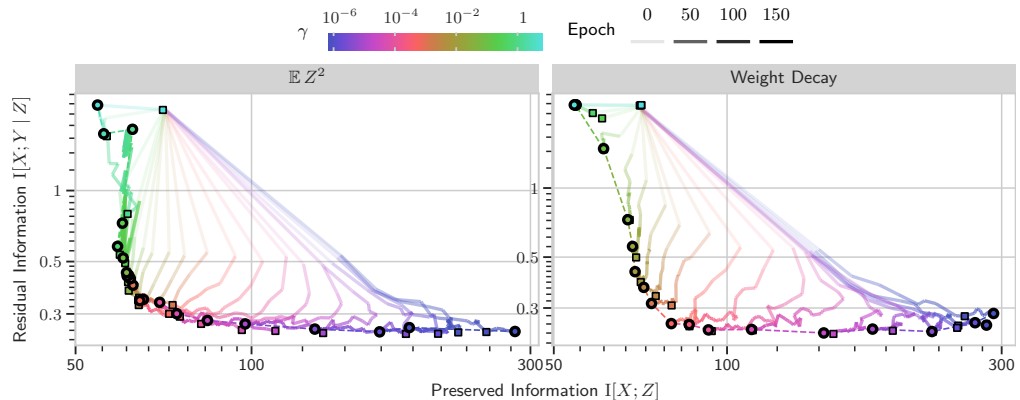

Figure 6: *Information plane plot of the training trajectories of ResNet18 models with the $\mathbb{E}\,\|Z\|^2$ surrogate objective or L2 weight-decay on CIFAR-10. The color shows $\gamma$; the transparency the training epoch. Compression (Preserved Information ↓) trades-off with performance (Residual Information ↓). See section 4. While the trajectories are similar, robustness is very different, see figure 5.*

**Information plane plots for CIFAR-10** To compare the different surrogate regularizers, we again use a ResNet18 model on CIFAR-10 with zero-entropy noise added to the final layer activations Z, with $K = 256$ dimensions, as an encoder and add a single $K \times 10$ linear unit as a decoder. We train with the surrogate objectives from section 3.3 for various $\gamma$, chosen in logspace from different ranges to compensate for their relationship to $\beta$ as noted in section 3.3: for $\log \mathrm{Var}[Z]$, $\gamma \in [10^{-5}, 1]$; for $\log \mathrm{Var}[Z \mid Y]$, $\gamma \in [10^{-5}, 10]$; and for $\mathbb{E}\,\|Z\|^2$, by trial and error, $\gamma \in [10^{-6}, 10]$. We estimate information quantities using the method of Kraskov et al. (2004).

Figure 6 shows an information plane plot for regularizing with $\mathbb{E}\,\|Z\|^2$ for different $\gamma$ over different epochs for the training set. Similar to Shwartz-Ziv and Tishby (2017), we observe that there is an initial expansion phase followed by compression. The jumps in performance (reduction of the Residual Information) are due to drops in the learning rate. In figure 4, we can see that the saturation curves for all 3 surrogate objectives qualitatively match the predicted curve from Tishby and Zaslavsky (2015). Figure G.1 shows the difference between the regularizers more clearly, and figure G.3 shows the training trajectories for all three regularizers. More details in section G.3.1.

**Information plane plots for Imagenette** To show that our surrogate objectives also scale up to larger datasets, we run a similar experiment on Imagenette (Howard, 2019), which is a subset of ImageNet with 10 classes with $224 \times 224 \times 3 = 1.5 \times 10^5$ input dimensions, and on which we obtain 90% test accuracy. See the figure 1, which shows the trajectories on the test set. We obtain similar plots to the ones obtained for CIFAR-10, *showing that our surrogate objectives scale well to higher-dimensional datasets despite their simplicity.*

## 5 CONCLUSION

The contributions of this paper have been threefold: First, we have proposed simple, tractable training objectives which capture many of the desirable properties of IB methods while also scaling to problems of interest in deep learning. For this we have introduced implicit stochastic encoders, e.g. using dropout, and compared multi-sample dropout approaches to identify the one that approximates the Decoder Uncertainty $\mathrm{H}_\theta[Y \mid Z]$, relating them to the cross-entropy loss that is commonly used for classification problems. This widens the range of DNN architectures that can be used with IB objectives considerably. We have demonstrated that our objectives perform well for practical DNNs without cumbersome density models. Second, we have motivated our objectives by providing insight into limitations of IB training, demonstrating how to avoid pathological behavior in IB objectives, and by endeavouring to provide a unifying view on IB approaches. Third, we have provided mathematically grounded intuition by using I-diagrams for the information quantities involved in IB, shown common pitfalls when using information quantities and how to avoid them, and examined how the quantities relate to each other. Future work investigating the practical constraints on the expressivity of a given neural network may provide further insight into how to measure compression in neural networks. Moreover, the connection to Bayesian Neural Networks remains to be explored.

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

## A    Information quantities & information diagrams

Here we introduce notation and terminology in greater detail than in the main paper. We review well-known information quantities and provide more details on using information diagrams (Yeung, 1991).

### A.1    Information quantities

We denote entropy $H[\cdot]$, joint entropy $H[\cdot, \cdot]$, conditional entropy $H[\cdot \mid \cdot]$, mutual information $I[\cdot; \cdot]$ and Shannon's information content $h(\cdot)$ following Cover and Thomas (2012); MacKay (2003); Shannon (1948) :

$$h(x) = -\ln x$$
$$H[X] = \mathbb{E}_{p(x)} h(p(x))$$
$$H[X, Y] = \mathbb{E}_{p(x,y)} h(p(x, y))$$
$$H[X \mid Y] = H[X, Y] - H[Y]$$
$$= \mathbb{E}_{p(y)} H[X \mid y] = \mathbb{E}_{p(x,y)} h(p(x \mid y))$$
$$I[X; Y] = H[X] + H[Y] - H[X, Y]$$
$$= \mathbb{E}_{p(x,y)} h\left(\tfrac{p(x)\,p(y)}{p(x,y)}\right)$$
$$I[X; Y \mid Z] = H[X \mid Z] + H[Y \mid Z] - H[X, Y \mid Z],$$

where $X, Y, Z$ are random variables and $x, y, z$ are outcomes these random variables can take.

We use differential entropies interchangeably with entropies. We can do so because equalities between them hold as can be verified by symbolic expansions. For example,

$$H[X, Y] = H[X \mid Y] + H[Y]$$
$$\Leftrightarrow \mathbb{E}_{p(x,y)} h(p(x, y)) = \mathbb{E}_{p(x,y)} [h(p(x \mid y)) + h(p(y))] = \mathbb{E}_{p(x,y)} [h(p(x \mid y))] + \mathbb{E}_{p(y)} h(p(y)),$$

which is valid in both the discrete and continuous case (if the integrals all exist). The question of how to transfer inequalities in the discrete case to the continuous case is dealt with in section 3.3.

We will further require the Kullback-Leibler divergence $D_{\mathrm{KL}}(\cdot \parallel \cdot)$ and cross-entropy $H(\cdot \parallel \cdot)$:

$$H(p(x) \parallel q(x)) = \mathbb{E}_{p(x)} h(q(x))$$
$$D_{\mathrm{KL}}(p(x) \parallel q(x)) = \mathbb{E}_{p(x)} h\left(\tfrac{q(x)}{p(x)}\right)$$
$$H(p(y \mid x) \parallel q(y \mid x)) = \mathbb{E}_{p(x)} \mathbb{E}_{p(y|x)} h(q(y \mid x))$$
$$= \mathbb{E}_{p(x,y)} h(q(y \mid x))$$
$$D_{\mathrm{KL}}(p(y \mid x) \parallel q(y \mid x)) = \mathbb{E}_{p(x,y)} h\left(\tfrac{q(y|x)}{p(y|x)}\right)$$

### A.2    Information diagrams

Information diagrams (I-diagrams), like the one depicted in figure 2 (or figure H.1 for a bigger version), visualize the relationship between information quantities: Yeung (1991) shows that we can define a signed measure $\mu^*$ such that these well-known quantities map to abstract sets and are consistent with set operations.

$$H[A] = \mu^*(A)$$
$$H[A_1, \ldots, A_n] = \mu^*(\cup_i A_i)$$
$$H[A_1, \ldots, A_n \mid B_1, \ldots, B_n] = \mu^*(\cup_i A_i - \cup_i B_i)$$
$$I[A_1; \ldots; A_n] = \mu^*(\cap_i A_i)$$
$$I[A_1; \ldots; A_n \mid B_1, \ldots, B_n] = \mu^*(\cap_i A_i - \cup_i B_i)$$

Note that interaction information (McGill, 1954) follows as canonical generalization of the mutual information to multiple variables from that work, whereas total correlation does not.

In other words, equalities can be read off directly from I-diagrams: an information quantity is the sum of its parts in the corresponding I-diagram. This is similar to Venn diagrams. The sets used in I-diagrams are just abstract symbolic objects, however.

An important distinction between I-diagrams and Venn diagrams is that while we can always read off inequalities in Venn diagrams, this is not true for I-diagrams in general because mutual information terms in more than two variables can be negative. In Venn diagrams, a set is always larger or equal any subset.

However, if we show that all information quantities are non-negative, we can read off inequalities again. We do this for figure 2 at the end of section 2 for categorical $Z$ and expand this to continuous $Z$ in section 3.3. Thus, we can treat the Mickey Mouse I-diagram like a Venn diagram to read off equalities and inequalities.

Nevertheless, caution is warranted sometimes. As the signed measure can be negative, $\mu^*(X \cap Y) = 0$ does *not* imply $X \cap Y = \emptyset$: deducing that a mutual information term is 0 does not imply that one can simply remove the corresponding area in the I-diagram. There could be $Z$ with $\mu^*((X \cap Y) \cap Z) < 0$, such that $\mu^*(X \cap Y) = \mu^*(X \cap Y \cap Z) + \mu^*(X \cap Y - Z) = 0$ but $X \cap Y \neq \emptyset$. This also means that we cannot drop the term from expressions when performing symbolic manipulations. This is of particular importance because a mutual information of zero means two random variables are independent, which might invite one drawing them as disjoint areas.

The only time where one can safely remove an area from the diagram is for *atomic* quantities, which are quantities which reference all the available random variables (Yeung, 1991). For example, when we only have three variables $X, Y, Z$, $I[X; Y; Z]$ and $I[X; Y \mid Z]$ are atomic quantities. We can safely remove atomic quantities from I-diagrams when they are 0 as there are no random variables left to apply that could lead to the problem explored above.

Continuing the example, $0 = I[X; Y; Z] = \mu^*(X \cap Y \cap Z)$ would imply $X \cap Y \cap Z = \emptyset$, and we could remove it from the diagram without loss of generality. Moreover, atomic $I[X; Y \mid Z] = \mu^*(X \cap Y - Z) = 0$ then and could be removed from the diagram as well.

We only use I-diagrams for the three variable case, but they supply us with tools to easily come up with equalities and inequalities for information quantities. In the general case with multiple variables, they can be difficult to draw, but for Markov chains they can be of great use.

# B    MICKEY MOUSE I-DIAGRAM

## B.1    INTUITION FOR THE MICKEY MOUSE INFORMATION QUANTITIES

We base the names of information quantities on existing conventions and come up with sensible extensions. For example, the name Preserved Relevant Information for $I[Y; Z]$ was introduced by Tishby and Zaslavsky (2015). It can be seen as the intersection of $I[X; Z]$ and $I[X; Y]$ in the I-diagram, and hence we denote $I[X; Z]$ Preserved Information and $I[X; Y]$ Relevant Information, which are sensible names as we detail below.

We identify the following six atomic quantities:

**Label Uncertainty** $H[Y \mid X]$ quantifies the uncertainty in our labels. If we have multiple labels for the same data sample, it will be $> 0$. It is 0 otherwise.

**Encoding Uncertainty** $H[Z \mid X]$ quantifies the uncertainty in our latent encoding given a sample. When using a Bayesian model with random variable $\omega$ for the weights, one can further split this term into $H[Z \mid X] = I[Z; \omega \mid X] + H[Z \mid X, \omega]$, so uncertainty stemming from weight uncertainty and independent noise (Houlsby et al., 2011; Kirsch et al., 2019).

**Preserved Relevant Information** $I[Y; Z]$ quantifies information in the latent that is relevant for our task of predicting the labels (Tishby and Zaslavsky, 2015). Intuitively, we want to maximize it for good predictive performance.

**Residual Information** $I[X; Y \mid Z]$ quantifies information for the labels that is not captured by the latent (Tishby and Zaslavsky, 2015) but would be useful to be captured.

**Redundant Information** $I[X; Z \mid Y]$ quantifies information in the latent that is not needed for predicting the labels[6].

We also identify the following composite information quantities:

**Relevant Information** $I[X; Y] = I[X; Y \mid Z] + I[Y; Z]$ quantifies the information in the data that is relevant for the labels and which our model needs to capture to be able to predict the labels.

**Preserved Information** $I[X; Z] = I[X; Z \mid Y] + I[Y; Z]$ quantifies information from the data that is preserved in the latent.

**Decoder Uncertainty** $H[Y \mid Z] = I[X; Y \mid Z] + H[Y \mid X]$ quantifies the uncertainty about the labels after learning about the latent Z. If $H[Y \mid Z]$ reaches 0, it means that no additional information is needed to infer the correct label Y from the latent Z: the optimal decoder can be a deterministic mapping. Intuitively, we want to minimize this quantity for good predictive performance.

**Reverse Decoder Uncertainty** $H[Z \mid Y] = I[X; Z \mid Y] + H[Z \mid X]$ quantifies the uncertainty about the latent Z given the label Y. We can imagine training a new model to predict Z given Y and minimizing $H[Z \mid Y]$ to 0 would allow for a deterministic decoder from the latent to given the label.

**Nuisance**[7] $H[X \mid Y] = H[X \mid Y, Z] + I[X; Z]$ quantifies the information in the data that is not relevant for the task (Achille and Soatto, 2018a).

### B.2 Definitions & equivalences

The following equalities can be read off from figure 2. For completeness and to provide a handy reference, we list them explicitly here. They can also be verified using symbolic manipulations and the properties of information quantities.

Equalities for composite quantities:

$$I[X; Y] = I[X; Y \mid Z] + I[Y; Z] \tag{17}$$
$$I[X; Z] = I[X; Z \mid Y] + I[Y; Z] \tag{18}$$
$$H[Y \mid Z] = I[X; Y \mid Z] + H[Y \mid X] \tag{19}$$
$$H[Z \mid Y] = I[X; Z \mid Y] + H[Z \mid X] \tag{20}$$
$$H[X \mid Y] = H[X \mid Y, Z] + I[X; Z] \tag{21}$$

We can combine the atomic quantities into the overall Label Entropy and Encoding Entropy:

$$H[Y] = H[Y \mid X] + I[Y; Z] + I[X; Y \mid Z] \tag{22}$$
$$H[Z] = H[Z \mid X] + I[Y; Z] + I[X; Z \mid Y]. \tag{23}$$

We can express the Relevant Information $I[X; Y]$, Residual Information $I[X; Y \mid Z]$, Redundant Information $I[X; Z \mid Y]$ and Preserved Information $I[X; Z]$ without X on the left-hand side:

$$I[X; Y] = H[Y] - H[Y \mid X], \tag{24}$$
$$I[X; Z] = H[Z] - H[Z \mid X], \tag{25}$$
$$I[X; Y \mid Z] = H[Y \mid Z] - H[Y \mid X], \tag{26}$$
$$I[X; Z \mid Y] = H[Z \mid Y] - H[Z \mid X]. \tag{27}$$

This simplifies estimating these expressions as X is usually much higher-dimensional and irregular than the labels or latent encodings. We also can rewrite the Preserved Relevant Information $I[Y; Z]$ as:

$$I[Y; Z] = H[Y] - H[Y \mid Z] \tag{28}$$
$$I[Y; Z] = H[Z] - H[Z \mid Y] \tag{29}$$

---

[6]Fisher (2019) uses the term "Residual Information" for this, which conflicts with Tishby and Zaslavsky (2015).

[7]Not depicted in figure 2.

## C    Information bottleneck & related works

### C.1    Goals & motivation

The IB principle from Tishby et al. (2000) can be recast as a generalization of finding minimal sufficient statistics for the labels given the data (Shamir et al., 2010; Tishby and Zaslavsky, 2015; Fisher, 2019): it strives for minimality and sufficiency of the latent Z. Minimality is about minimizing amount of information necessary of X for the task, so minimizing the Preserved Information $I[X; Z]$; while sufficiency is about preserving the information to solve the task, so maximizing the Preserved Relevant Information $I[Y; Z]$.

From figure 2, we can read off the definitions of Relevant Information and Preserved Information:

$$I[X; Y] = I[Y; Z] + I[X; Y | Z] \tag{30}$$

$$I[X; Z] = I[Y; Z] + I[X; Z | Y], \tag{31}$$

and see that maximizing the Preserved Relevant Information $I[Y; Z]$ is equivalent to minimizing the Residual Information $I[X; Y | Z]$, while minimizing the Preserved Information $I[X; Z]$ at the same time means minimizing the Redundant Information $I[X; Z | Y]$, too, as $I[X; Y]$ is constant for the given dataset[8]. Moreover, we also see that the Preserved Relevant Information $I[Y; Z]$ is upper-bounded by Relevant Information $I[X; Y]$, so to capture all relevant information in our latent, we want $I[X; Y] = I[Y; Z]$.

Using the diagram, we can also see that minimizing the Residual Information is the same as minimizing the Decoder Uncertainty $H[Y | Z]$:

$$I[X; Y | Z] = H[Y | Z] - H[Y | X].$$

Ideally, we also want to minimize the Encoding Uncertainty $H[Z | X]$ to find the most deterministic latent encoding Z. Minimizing the Encoding Uncertainty and the Redundant Information $I[X; Z | Y]$ together is the same as minimizing the Reverse Decoder Uncertainty $H[Z | Y]$.

All in all, we want to minimize both the Decoder Uncertainty $H[Y | Z]$ and the Reverse Decoder Uncertainty $H[Z | Y]$.

### C.2    IB objectives

**"The Information Bottleneck Method" (IB)**

Tishby et al. (2000) introduce $MI(X; \hat{X}) - \beta MI(\hat{X}; Y)$ as optimization objective for the Information Bottleneck. We can relate this to our notation by renaming $\hat{X} = Z$, such that the objective becomes "min $I[X; Z] - \beta I[Y; Z]$". The IB objective minimizes the Preserved Information $I[X; Z]$ and trades it off with maximizing the Preserved Relevant Information $I[Y; Z]$. Tishby and Zaslavsky (2015) mention that the IB objective is equivalent to minimizing $I[X; Z] + \beta I[X; Y | Z]$, see our discussion above. Tishby et al. (2000) provide an optimal algorithm for the tabular case, when X, Y and Z are all categorical. This has spawned additional research to optimize the objective for other cases and specifically for DNNs.

**"Deterministic Information Bottleneck" (DIB)**

Strouse and Schwab (2017) introduce as objective "min $H[Z] - \beta I[Y; Z]$". Compared to the IB objective, this also minimizes $H[Z | X]$ and encourages determinism. Vice-versa, for deterministic encoders, $H[Z | X] = 0$, and their objective matches the IB objective. Like Tishby et al. (2000), they provide an algorithm for the tabular case. To do so, they examine an analytical solution for their objective as it is unbounded: $H[Z | X] \rightarrow -\infty$ for the optimal solution. As we discuss in section 3.3, it does not easily translate to a continuous latent representation.

**"Deep Variational Information Bottleneck"**

Alemi et al. (2016) rewrite the terms in the bottleneck as maximization problem "max $I[Y; Z] - \beta I[X; Z]$" and swap the $\beta$ parameter. Their $\beta$ would be $1/\beta$ in IB above, which emphasizes that $I[Y; Z]$ is important for performance and $I[X; Z]$ acts as regularizer.

---

[8]That is, it does not depend on $\theta$.

The paper derives the following variational approximation to the IB objective, where $z = f_\theta(x, \epsilon)$ denotes a stochastic latent embedding with distribution $p_\theta(z \mid x)$, $p_\theta(\hat{y} \mid z)$ denotes the decoder, and $r(z)$ is some fixed prior distribution on the latent embedding:

$$\min \mathbb{E}_{\hat{p}(x,y)} \mathbb{E}_{\epsilon \sim p(\epsilon)} \left[ -\log p_\theta(\hat{Y} = y \mid z = f_\theta(x_n, \epsilon)) + \gamma \, D_{\mathrm{KL}}(p(z|x_n) \| r(z)) \right]. \tag{32}$$

In principle, the distributions $p_\theta(\hat{y} \mid z)$ and $p_\theta(z \mid x)$ could be given by arbitrary parameterizations and function approximators. In practice, the implementation of DVIB presented by Alemi et al. (2016) constructs $p_\theta(z \mid x)$ as a multivariate Gaussian with parameterized mean and parameterized diagonal covariance using a neural network, and then uses a simple logistic regression to obtain $p_\theta(\hat{y} \mid z)$, while arbitrarily setting $r(z)$ to be a unit Gaussian around the origin. The requirement for $p_\theta(z \mid x)$ to have a closed-form Kullback-Leibler divergence limits the applicability of the DVIB objective.

The DVIB objective can be written more concisely as

$$\min H_\theta[Y \mid Z] + \gamma \, D_{\mathrm{KL}}(p(z \mid x) \| r(z))$$

in the notation introduced in section 3. We discuss the regularizer in more detail in section F.3.

"CONDITIONAL ENTROPY BOTTLENECK"

In a preprint, Fisher (2019) introduce their Conditional Entropy Bottleneck as "$\min I[X; Z \mid Y] - I[Y; Z]$". We can rewrite the objective as $I[X; Z \mid Y] + I[X; Y \mid Z] - I[X; Y]$, using equations (30) and (31). The last term is constant for the dataset and can thus be dropped. Likewise, the IB objective can be rewritten as minimizing $I[X; Z \mid Y] + (\beta - 1)I[X; Y \mid Z]$. The two match for $\beta = 2$. Fisher (2019) provides experimental results that favorably compare to Alemi et al. (2016), possibly due to additional flexibility as Fisher (2019) do not constrain $p(z)$ to be a unit Gaussian and employ variational approximations for all terms. We relate CEB to Entropy Distance Metric in section C.4.

"CONDITIONAL ENTROPY BOTTLENECK" (2020)

In a substantial revision of the preprint, Fischer (2020) change their Conditional Entropy Bottleneck to include a Lagrange multiplier: "$\min I[X; Z \mid Y] - \gamma I[Y; Z]$". Their VCEB objective can be written more concisely as

$$\min H_\theta[Y \mid Z] + \gamma(H_\theta[Z \mid Y] - H_\theta[Z \mid X]),$$

where, without writing down the probabilistic model, we introduce variational approximations for the Reverse Decoder Uncertainty and the Encoding Uncertainty.

They are the first to report results on CIFAR-10. It is not clear how they parameterize the model they use for CIFAR-10. They use one Gaussian per class to model $H_\theta[Z \mid Y]$.

"CEB IMPROVES MODEL ROBUSTNESS"

Fischer and Alemi (2020) take CEB and switch to a deterministic model which they turn it into a stochastic encoder by adding unit Gaussian noise. They use Gaussians of fixed variance to variationally approximate $q(y \mid z)$: for each class, $q(y \mid z)$ is modelled as a separate Gaussian.

They are the first to report results on ImageNet and report good rebustness against adversarial attacks without adversarial training.

## C.3  CANONICAL IB & DIB OBJECTIVES

We expand the IB and DIB objectives into "disjoint" terms and drop constant ones to find a more canonical form. This leads us to focus on the optimization of the Decoder Uncertainty $H[Y \mid Z]$ along with additional regularization terms. In section 3.2, we discuss the properties of $H[Y \mid Z]$, and in section 3.3 we examine the regularization terms.

**Proposition.** *For IB, we obtain*

$$\arg\min I[X; Z] - \beta I[Y; Z] = \arg\min H[Y \mid Z] + \beta' \underbrace{I[X; Z \mid Y]}_{= H[Z|Y] - H[Z|X]}, \tag{33}$$

*and, for DIB,*

$$\arg\min H[Z] - \beta I[Y; Z] = \arg\min H[Y \mid Z] + \beta' H[Z \mid Y] = \arg\min H[Y \mid Z] + \beta'' H[Z] \tag{34}$$

*with $\beta' := \frac{1}{\beta-1} \in [0, \infty)$ and $\beta'' := \frac{1}{\beta} \in [0, 1)$.*

*Proof.* For the steps marked with *, we make use of $\beta > 1$. For IB, we obtain

$$\arg\min I[X; Z] - \beta I[Y; Z] = \arg\min I[X; Z \mid Y] + (\beta - 1)H[Y \mid Z]$$

$$\overset{(*)}{=} \arg\min H[Y \mid Z] + \beta' \, I[X; Z \mid Y]$$

$$\arg\min H[Y \mid Z] + \beta'(H[Z \mid Y] - H[Z \mid X]), \tag{IB}$$

and, for DIB,

$$\arg\min H[Z] - \beta I[Y; Z] = \arg\min H[Z \mid Y] + (\beta - 1)H[Y \mid Z]$$

$$\overset{(*)}{=} \arg\min H[Y \mid Z] + \beta' H[Z \mid Y], \tag{DIB}$$

with $\beta' := \frac{1}{\beta-1} \in [0, \infty)$. Similarly, we show for DIB

$$\arg\min H[Z] - \beta I[Y; Z] = \arg\min H[Z] + \beta H[Y \mid Z]$$

$$\overset{(*)}{=} \arg\min H[Y \mid Z] + \beta'' H[Z],$$

with $\beta'' := \frac{1}{\beta} \in [0, 1)$, which is relevant in section 3.3.

We limit ourselves to $\beta > 1$, because, for $\beta < 1$, we would be maximizing the Decoder Uncertainty, which does not make sense: the obvious solution to this is one where Z contains no information on Y, that is $p(y \mid z)$ is uniform. In the case of DIB, it is to map every input deterministically to a single latent; whereas for IB, we only minimize the Redundant Information, and the solution is free to contain noise. For $\beta = 1$, we would not care about Decoder Uncertainty and only minimize Redundant Information and Reverse Decoder Uncertainty, respectively, which allows for arbitrarily bad predictions. $\square$

We note that we have $\beta' = \frac{\beta''}{1-\beta''}$ using the relations above.

## C.4 IB OBJECTIVES AND THE ENTROPY DISTANCE METRIC

Another perspective on the IB objectives is by expressing them using the Entropy Distance Metric. MacKay (2003, p. 140) introduces the entropy distance

$$EDM(Y, Z) = H[Y \mid Z] + H[Z \mid Y]. \tag{35}$$

as a metric when we identify random variables up to permutations of the labels for categorical variables: if the entropy distance is 0, Y and Z are the same distribution up to a consistent permutation of the labels (independent of X). If the entropy distance becomes 0, both $H[Y \mid Z] = 0 = H[Z \mid Y]$, and we can find a bijective map from Z to Y.[9]

We can express the Reverse Decoder Uncertainty $H[Z \mid Y]$ using the Decoder Uncertainty $H[Y \mid Z]$ and the entropies:

$$H[Z \mid Y] + H[Y] = H[Y \mid Z] + H[Z],$$

and rewrite equation (35) as

$$EDM(Y, Z) = 2H[Y \mid Z] + H[Z] - H[Y].$$

For optimization purposes, we can drop constant terms and rearrange:

$$\arg\min EDM(Y, Z) = \arg\min H[Y \mid Z] + \tfrac{1}{2}H[Z].$$

### C.4.1 REWRITING IB AND DIB USING THE ENTROPY DISTANCE METRIC

For $\beta \geq 1$, we can rewrite equations (IB) and (DIB) as:

$$\arg\min EDM(Y, Z) + \gamma(H[Y \mid Z] - H[Z \mid Y]) + (\gamma - 1)H[Z \mid X] \tag{36}$$

for IB, and

$$\arg\min EDM(Y, Z) + \gamma(H[Y \mid Z] - H[Z \mid Y]) \tag{37}$$

---

[9]The argument for continuous variables is the same. We need to identify distributions up to "isentropic" bijections.

for DIB and replace $\beta$ with $\gamma = 1 - \frac{2}{\beta} \in [-1, 1]$ which allows for a linear mix between $H[Y \mid Z]$ and $H[Z \mid Y]$.

DIB will encourage the model to match both distributions for $\gamma = 0$ ($\beta = 2$), as we obtain a term that matches the Entropy Distance Metric from section C.4, and otherwise trades off Decoder Uncertainty and Reverse Decoder Uncertainty. IB behaves similarly but tends to maximize Encoding Uncertainty as $\gamma - 1 \in [-2, 0]$. Fisher (2019) argues for picking this configuration similar to the arguments in section C.1. DIB will force both distributions to become exactly the same, which would turn the decoder into a permutation matrix for categorical variables.

# D    DECODER UNCERTAINTY $H[Y \mid Z]$

## D.1    CROSS-ENTROPY LOSS

The cross-entropy loss features prominently in section 3.2. We can derive the usual cross-entropy loss for our model by minimizing the Kullback-Leibler divergence between the empirical sample distribution $\hat{p}(x, y)$ and the parameterized distribution $p_\theta(x) \, p_\theta(\hat{y} \mid x)$. For discriminative models, we are only interested in $p_\theta(\hat{y} \mid x)$, and can simply set $p_\theta(\hat{y} \mid x) = \hat{p}(x)$:

$$\arg\min_\theta D_{\mathrm{KL}}(\hat{p}(x, y) \,\|\, p_\theta(x) \, p_\theta(\hat{Y} = y \mid x))$$

$$= \arg\min_\theta D_{\mathrm{KL}}(\hat{p}(y \mid x) \,\|\, p_\theta(\hat{Y} = y \mid x)) + \underbrace{D_{\mathrm{KL}}(\hat{p}(x) \,\|\, p_\theta(x))}_{=0}$$

$$= \arg\min_\theta H(\hat{p}(y \mid x) \,\|\, p_\theta(\hat{Y} = y \mid x)) - \underbrace{H[Y \mid X]}_{\text{const.}}$$

$$= \arg\min_\theta H(\hat{p}(y \mid x) \,\|\, p_\theta(\hat{Y} = y \mid x)).$$

In section 3.2, we introduce the shorthand $H_\theta[Y \mid X]$ for $H(\hat{p}(y \mid x) \,\|\, p_\theta(\hat{Y} = y \mid x))$ and refer to it as Prediction Cross-Entropy.

## D.2    UPPER BOUNDS & TRAINING ERROR MINIMIZATION

To motivate that $H[Y \mid Z]$ (or $H_\theta[Y \mid Z]$) can be used as main loss term, we show that it can bound the (training) error probability since *accuracy* is often the true objective when machine learning models are deployed on real-world problems[10].

**Proposition.** *The Decoder Cross-Entropy provides an upper bound on the Decoder Uncertainty:*

$$H[Y \mid Z] \le H[Y \mid Z] + D_{\mathrm{KL}}(p(y \mid z) \,\|\, p_\theta(\hat{y} \mid z)) = H_\theta[Y \mid Z],$$

*and further bounds the training error:*

$$p(\text{``}\hat{Y} \text{ is wrong''}) \le 1 - e^{-H_\theta[Y\mid Z]} = 1 - e^{-\left(H[Y\mid Z] + D_{\mathrm{KL}}(p(y\mid z)\|p_\theta(\hat{y}\mid z))\right)}.$$

*Likewise, for the Prediction Cross-Entropy $H_\theta[Y \mid X]$ and the Label Uncertainty $H[Y \mid X]$.*

*Proof.* The upper bounds for Decoder Uncertainty $H[Y \mid Z]$ and Label Uncertainty $H[Y \mid X]$ follow from the non-negativity of the Kullback-Leibler divergence, for example:

$$0 \le D_{\mathrm{KL}}(p(y \mid z) \,\|\, p_\theta(\hat{y} \mid z)) = H_\theta[Y \mid Z] - H[Y \mid Z],$$

$$0 \le D_{\mathrm{KL}}(\hat{p}(y \mid x) \,\|\, p_\theta(\hat{y} \mid x)) = H_\theta[Y \mid X] - H[Y \mid X].$$

The derivation for the training error probability is as follows:

$$p(\text{``}\hat{Y} \text{ is correct''}) = \mathbb{E}_{\hat{p}(x,y)} \, p(\text{``}\hat{Y} \text{ is correct''} \mid x, y) = \mathbb{E}_{\hat{p}(x,y)} \, \mathbb{E}_{p_\theta(z\mid x)} \, p_\theta(\hat{Y} = y \mid z)$$

$$= \mathbb{E}_{p(y,z)} \, p_\theta(\hat{Y} = y \mid z).$$

We can then apply Jensen's inequality using convex $h(x) = -\ln x$:

$$h\left(\mathbb{E}_{p(y,z)} \, p_\theta(\hat{Y} = y \mid z)\right) \le \mathbb{E}_{p(y,z)} \, h\left(p_\theta(\hat{Y} = y \mid z)\right)$$

---

[10]As we only take into account the empirical distribution $\hat{p}(x, y)$ available for training, the following derivation refers only to the empirical risk, and not to the expected risk of the estimator $\hat{Y}$.

$$\Leftrightarrow p(\text{``}\hat{Y}\text{ is correct''}) \geq e^{-H(p(y|z)\|p_\theta(\hat{Y}=y|z))}$$

$$\Leftrightarrow p(\text{``}\hat{Y}\text{ is wrong''}) \leq 1 - e^{-H_\theta[Y|Z]}.$$

For small $H_\theta[Y \mid Z]$, we note that one can use the approximation $e^x \approx 1 + x$ to obtain:

$$p(\text{``}\hat{Y}\text{ is wrong''}) \lessgtr H_\theta[Y \mid Z]. \tag{38}$$

Finally, we split the Decoder Cross-Entropy into the Decoder Uncertainty and a Kullback-Leibler divergence:

$$H_\theta[Y \mid Z] = H[Y \mid Z] + D_{KL}(p(y \mid z) \,\|\, p_\theta(\hat{Y} = y \mid z)).$$

If we upper-bound $D_{KL}(p(y|z)\|p_\theta(\hat{Y}=y|z))$, minimizing the Decoder Uncertainty $H[Y \mid Z]$ becomes a sensible minimization objective as it reduces the probability of misclassification.

We can similarly show that the training error is bounded by the Prediction Cross-Entropy $H_\theta[Y \mid X]$. $\qquad\square$

In the next section, we examine categorical Z for which optimal decoders can be constructed and $D_{KL}(p(y \mid z) \,\|\, p_\theta(\hat{Y} = y \mid z))$ becomes zero.

# E    CATEGORICAL Z

For categorical Z, $p(y \mid z)$ can be computed exactly for a given encoder $p_\theta(z \mid x)$ by using the empirical data distribution, which, in turn, allows us to compute $H[Y \mid Z]$[11]. This is similar to computing a confusion matrix between Y and Z but using information content instead of probabilities.

Moreover, if we set $p_\theta(\hat{y} \mid z) := p(Y = \hat{y} \mid z)$ to have an optimal decoder, we obtain equality in equation (9), and obtain $H_\theta[Y \mid X] \leq H_\theta[Y \mid Z] = H[Y \mid Z]$. If the encoder were also deterministic, we would obtain $H_\theta[Y \mid X] = H_\theta[Y \mid Z] = H[Y \mid Z]$. We can minimize $H[Y \mid Z]$ directly using gradient descent. $\frac{d}{d\theta}H[Y \mid Z]$ only depends on $p(y \mid z)$ and $\frac{d}{d\theta} p_\theta(z \mid x)$:

$$\frac{d}{d\theta}H[Y \mid Z] = \mathbb{E}_{p(x,z)} \left[ \frac{d}{d\theta} \left[ \ln p_\theta(z \mid x) \right] \mathbb{E}_{\hat{p}(y|x)} h\left(p(y \mid z)\right) \right].$$

*Proof.*

$$\frac{d}{d\theta}H[Y \mid Z] = \frac{d}{d\theta} \mathbb{E}_{p(y,z)} h\left(p(y \mid z)\right) = \frac{d}{d\theta} \mathbb{E}_{p(x,y,z)} h\left(p(y \mid z)\right) = \mathbb{E}_{\hat{p}(x,y)} \frac{d}{d\theta} \mathbb{E}_{p_\theta(z|x)} h\left(p(y \mid z)\right)$$

$$= \mathbb{E}_{p_\theta(z|x)} \mathbb{E}_{\hat{p}(x,y)} \frac{d}{d\theta} \left[ h\left(p(y \mid z)\right) \right] + h\left(p(y \mid z)\right) \frac{d}{d\theta} \left[ \ln p_\theta(z \mid x) \right]$$

$$= \mathbb{E}_{p(x,y,z)} \frac{d}{d\theta} \left[ h\left(p(y \mid z)\right) \right] + h\left(p(y \mid z)\right) \frac{d}{d\theta} \left[ \ln p_\theta(z \mid x) \right].$$

And now we show that $\mathbb{E}_{p(x,y,z)} \frac{d}{d\theta} \left[ h\left(p(y \mid z)\right) \right] = 0$:

$$\mathbb{E}_{p(x,y,z)} \frac{d}{d\theta} \left[ h\left(p(y \mid z)\right) \right] = \mathbb{E}_{p(y,z)} \frac{d}{d\theta} \left[ h\left(p(y \mid z)\right) \right] = \mathbb{E}_{p(y,z)} \frac{-1}{p(y \mid z)} \frac{d}{d\theta} p(y \mid z)$$

$$= -\int \frac{p(y, z)}{p(y \mid z)} \frac{d}{d\theta} p(y \mid z) \, dy \, dz = -\int p(z) \int \frac{d}{d\theta} p(y \mid z) \, dy \, dz$$

$$= -\int p(z) \frac{d}{d\theta} \Big[ \underbrace{\int p(y \mid z) \, dy}_{=1} \Big] dz = 0.$$

Splitting the expectation and reordering of $\mathbb{E}_{p(x,y,z)} h\left(p(y \mid z)\right) \frac{d}{d\theta} \left[ \ln p_\theta(z \mid x) \right]$, we obtain the result. $\quad\square$

The same holds for Reverse Decoder Uncertainty $H[Z \mid Y]$ and for the other quantities as can be verified easily.

If we minimize $H[Y \mid Z]$ directly, we can compute $p(y \mid z)$ after every training epoch and fix $p_\theta(\hat{y} \mid z) := p(Y = \hat{y} \mid z)$ to create the discriminative model $p_\theta(\hat{y} \mid x)$. This is a different perspective on the self-consistent equations from Tishby et al. (2000); Gondek and Hofmann (2003).

---

[11]$p(y \mid z)$ depends on $\theta$ through $p_\theta(z \mid x)$: $p(y \mid z) = \frac{\sum_x \hat{p}(x,y) p_\theta(z|x)}{\sum_x \hat{p}(x) p_\theta(z|x)}$.

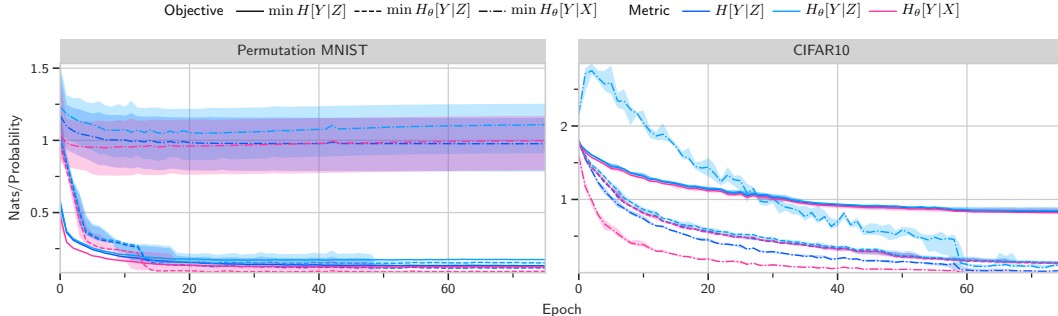

Figure E.1: *Decoder Uncertainty, Decoder Cross-Entropy and Prediction Cross-Entropy for Permutation-MNIST and CIFAR-10 with a categorical Z.* $C = 100$ *categories are used for Z. We optimize with different minimization objectives in turn and plot the metrics.* $D_{KL}(p(y \mid z) \parallel p_\theta(\hat{Y} = y \mid z))$ *is small when training with* $H_\theta[Y \mid Z]$ *or* $H[Y \mid Z]$. *When training with* $H_\theta[Y \mid X]$ *on CIFAR-10,* $D_{KL}(p(y \mid z) \parallel p_\theta(\hat{Y} = y \mid z))$ *remains quite large. We run 8 trials each and plot the median with confidence bounds (25% and 75% quartiles). See section E.1 for more details.*

### E.1 EMPIRICAL EVALUATION OF $D_{KL}(p(y \mid z) \parallel p_\theta(\hat{Y} = y \mid z))$ DURING TRAINING

We examine the size of the gap between Decoder Uncertainty and Decoder Cross-Entropy and the training behavior of the two cross-entropies with *categorical* latent Z on Permutation MNIST and CIFAR-10. For Permutation MNIST (Goodfellow et al., 2013), we use the common fully-connected ReLU $784 - 1024 - 1024 - C$ encoder architecture, with $C = 100$ categories for Z. For CIFAR-10 (Krizhevsky et al., 2009), we use a standard ResNet18 model with $C$ many output classes as encoder (He et al., 2016a). See section G for more details about the hyperparameters. Even though a $C \times 10$ matrix and a SoftMax would suffice to describe the decoder matrix $p_\theta(\hat{y} \mid z)$[12], we have found that over-parameterization using a separate DNN benefits optimization a lot. Thus, to parameterize the decoder matrix, we use fully-connected ReLUs $C - 1024 - 1024 - 10$ with a final SoftMax layer. We compute it once per batch during training and back-propagate into it.

Figure E.1 shows the three metrics as we train with each of them in turn. Our results do not achieve SOTA accuracy on the test set—we impose a harder optimization problem as Z is categorical, and we are essentially solving a hard-clustering problem first and then map these clusters to $\hat{Y}$. Results are provided for the training set in order to compare with the optimal decoder.

As predicted, the Decoder Cross-Entropy upper-bounds both the Decoder Uncertainty $H[Y \mid Z]$ and the Prediction Cross-Entropy in all cases. Likewise, the gap between $H_\theta[Y \mid Z]$ and $H[Y \mid Z]$ is tiny when we minimize $H_\theta[Y \mid Z]$. On the other hand, minimizing Prediction Cross-Entropy can lead to large gaps between $H_\theta[Y \mid Z]$ and $H[Y \mid Z]$, as can be seen for CIFAR-10.

Very interestingly, on MNIST Decoder Cross-Entropy provides a better training objective whereas on CIFAR-10 Prediction Cross-Entropy trains lower. Decoder Uncertainty does not train very well on CIFAR-10, and Prediction Cross-Entropy does not train well on Permutation MNIST at all. We suspect DNN architectures in the literature have evolved to train well with cross-entropies, but we are surprised by the heterogeneity of the results for the two datasets and models.

## F SURROGATES FOR REGULARIZATION TERMS

### F.1 DIFFERENTIAL ENTROPIES

**Proposition.** *After adding zero-entropy noise, the inequality* $I[X; Z \mid Y] \leq H[Z \mid Y] \leq H[Z]$ *also holds in the continuous case, and we can minimize* $I[X; Z \mid Y]$ *in the IB objective by minimizing* $H[Z \mid Y]$ *or* $H[Z]$, *similarly to the DIB objective. We present a formal proof in section F.1.*

**Theorem 2.** *For random variables A, B, we have*

$$H[A + B] \geq H[B].$$

---

[12]For categorical Z, $p_\theta(\hat{y} \mid z)$ is a stochastic matrix which sums to 1 along the $\hat{Y}$ dimension.

*Proof.* See Bercher and Vignat (2002, section 2.2). □

**Proposition 1.** *Let Y, Z and X be random variables satisfying the independence property $Z \perp Y|X$, and F a possibly stochastic function such that $Z = F(X) + \epsilon$, with independent noise $\epsilon$ satisfying $\epsilon \perp F(X), \epsilon \perp Y$ and $H(\epsilon) = 0$. Then the following holds whenever $I[Y; Z]$ is well-defined.*

$$I[X; Z \mid Y] \le H[Z \mid Y] \le H[Z].$$

*Proof.* First, we note that $H[Z \mid X] = H[F(X) + \epsilon \mid X] \ge H[\epsilon \mid X] = H[\epsilon]$ with theorem 2, as $\epsilon$ is independent of $X$, and thus $H[Z \mid X] \ge 0$. We have $H[Z \mid X] = H[Z \mid X, Y]$ by the conditional independence assumption, and by the non-negativity of mutual information, $I[Y; Z] \ge 0$. Then:

$$I[X; Z \mid Y] + \underbrace{H[Z \mid X]}_{\ge 0} = H[Z \mid Y]$$

$$H[Z \mid Y] + \underbrace{I[Y; Z]}_{\ge 0} = H[Z]$$

□

The probabilistic model from section 2 fulfills the conditions exactly, and the two statements motivate our proposition.

It is important to note that while zero-entropy noise is necessary for preserving inequalities like $I[X; Z \mid Y] \le H[Z \mid Y] \le H[Z]$ in the continuous case, any Gaussian noise will suffice for optimization purposes: we optimize via pushing down an upper bound, and constant offsets will not affect this.

Thus, if we had $H[\epsilon] \ne 0$, even though $I[X; Z \mid Y] + H[Z \mid X] \nleq H[Z \mid Y]$, we could instead use

$$I[X; Z \mid Y] + H[Z \mid X] - H[\epsilon] \le H[Z \mid Y] - H[\epsilon]$$

as upper bound to minimize. The gradients remain the same.

This also points to the nature of differential entropies as lacking a proper point of origin by themselves. We choose one by fixing $H[\epsilon]$. Just like other literature usually only considers mutual information as meaningful, we consider $H[Z \mid X] - H[\epsilon]$ as more meaningful than $H[Z \mid X]$. However, we can side-step this discussion conveniently by picking a canonical noise as point of origin in the form of zero-entropy noise $H[\epsilon] = 0$.

### F.2 UPPER BOUNDS

We derive this result as follows:

$$\begin{aligned}
H[Z \mid Y] &= \mathbb{E}_{\hat{p}(y)} H[Z \mid y] \\
&\le \mathbb{E}_{\hat{p}(y)} \tfrac{1}{2} \ln \det(2\pi e \, \mathrm{Cov}[Z \mid y]) \\
&\le \mathbb{E}_{\hat{p}(y)} \sum_i \tfrac{1}{2} \ln(2\pi e \, \mathrm{Var}[Z_i \mid y]) \\
&\approx \mathbb{E}_{\hat{p}(y)} \sum_i \tfrac{1}{2} \ln(2\pi e \, \widehat{\mathrm{Var}}[Z_i \mid y]),
\end{aligned}$$

**Theorem 3.** *Given a k-dimensional random variable $X = (X_i)_{i=1}^k$ with $\mathrm{Var}[X_i] > 0$ for all i,*

$$H[X] \le \tfrac{1}{2} \ln \det(2\pi e \, \mathrm{Cov}[X])$$
$$\le \sum_i \tfrac{1}{2} \ln(2\pi e \, \mathrm{Var}[X_i]).$$

*Proof.* First, the multivariate normal distribution with same covariance is the maximum entropy distribution for that covariance, and thus $H[X] \le \ln \det(2\pi e \, \mathrm{Cov}[X])$, when we substitute the differential entropy for a multivariate normal distribution with covariance $\mathrm{Cov}[X]$. Let $\Sigma_0 := \mathrm{Cov}[X]$ be the covariance matrix and $\Sigma_1 := \mathrm{diag}(\mathrm{Var}[X_i])_i$ the matrix that only contains the diagonal. Because we add independent noise, $\mathrm{Var}[X_i] > 0$ and thus $\Sigma_1^{-1}$ exists. It is clear that $\mathrm{tr}(\Sigma_1^{-1}\Sigma_0) = k$. Then, we can use the KL-Divergence between two multivariate normal distributions $\mathcal{N}_0, \mathcal{N}_1$ with same mean 0 and

covariances $\Sigma_0$ and $\Sigma_1$ to show that $\ln \det \Sigma_0 \leq \ln \det \Sigma_1$:

$$0 \leq D_{\mathrm{KL}}(\mathcal{N}_0 \| \mathcal{N}_1) = \tfrac{1}{2}\left(\mathrm{tr}(\Sigma_1^{-1}\Sigma_0) - k + \ln\left(\frac{\det \Sigma_1}{\det \Sigma_0}\right)\right)$$

$$\Leftrightarrow 0 \leq \tfrac{1}{2}\ln\left(\frac{\det \Sigma_1}{\det \Sigma_0}\right) \Leftrightarrow \tfrac{1}{2}\ln \det \Sigma_0 \leq \tfrac{1}{2}\ln \det \Sigma_1.$$

We substitute the definitions of $\Sigma_0$ and $\Sigma_1$, and obtain the second inequality after adding $k\ln(2\pi e)$ on both sides. □

**Theorem 4.** *Given a k-dimensional real-valued random variable $X = (X_i)_{i=1}^{k} \in \mathbb{R}^k$, we can bound the entropy by the mean squared norm of the latent:*

$$\mathbb{E}\,\|X\|^2 \leq C' \Rightarrow \mathrm{H}[X] \leq C, \tag{39}$$

*with $C' := \frac{ke^{2C/k}}{2\pi e}$.*

*Proof.* We begin with the previous bound:

$$\mathrm{H}[X] \leq \sum_i \tfrac{1}{2}\ln(2\pi e\,\mathrm{Var}[X_i]) = \tfrac{k}{2}\ln 2\pi e + \tfrac{1}{2}\ln\prod_i \mathrm{Var}[X_i]$$

$$\leq \tfrac{k}{2}\ln 2\pi e + \tfrac{1}{2}\ln\left(\tfrac{1}{k}\sum_i \mathrm{Var}[X_i]\right)^k = \tfrac{k}{2}\ln\tfrac{2\pi e}{k}\sum_i \mathrm{Var}[X_i]$$

$$\leq \tfrac{k}{2}\ln\tfrac{2\pi e}{k}\,\mathbb{E}\,\|X\|^2,$$

where we use the AM-GM inequality:

$$\left(\prod_i \mathrm{Var}[X_i]\right)^{\frac{1}{k}} \leq \tfrac{1}{k}\sum_i \mathrm{Var}[X_i]$$

and the monotony of the logarithm with:

$$\sum_i \mathrm{Var}[X_i] = \sum_i \mathbb{E}\left[X_i^2\right] - \mathbb{E}\left[X_i\right]^2 \leq \sum_i \mathbb{E}\left[X_i^2\right] = \mathbb{E}\,\|X\|^2$$

Bounding using $\mathbb{E}\,\|X\|^2 \leq C'$, we obtain

$$\mathrm{H}[X] \leq \tfrac{k}{2}\ln\tfrac{2\pi e}{k}C' = C,$$

and solving for $C'$ yields the statement. □

This theorem provides justification for the use of $\ln \mathbb{E}\,\|Z\|^2$ as a regularizer, but does not justify the use of $\mathbb{E}\,\|Z\|^2$ directly. Here, we give two motivations. We first observe that $\ln x \leq x - 1$ due to $\ln$'s strict convexity and $\ln 1 = 0$, and thus:

$$\mathrm{H}[X] \leq \tfrac{k}{2}\ln\tfrac{2\pi e}{k}\,\mathbb{E}\,\|X\|^2 = \tfrac{k}{2}\left(\ln\tfrac{2\pi}{k}\,\mathbb{E}\,\|X\|^2 - 1\right) \leq \pi\,\mathbb{E}\,\|X\|^2.$$

We can also take a step back and remind ourselves that IB objectives are actually Lagrangians, and $\beta$ in $\min \mathrm{I}[X;Z] - \beta\mathrm{I}[Y;Z]$ is introduced as Lagrangian multiplier for the constrained objective:

$$\min \mathrm{I}[X;Z] \text{ s.t. } \mathrm{I}[Y;Z] \geq C.$$

We can similarly write our canonical DIB objective $\mathrm{H}[Y\,|\,Z] + \beta''\mathrm{H}[Z]$ as constrained objective

$$\min \mathrm{H}[Y\,|\,Z] \text{ s.t. } \mathrm{H}[Z] \leq C,$$

and use above statement to find the approximate form

$$\min \mathrm{H}[Y\,|\,Z] \text{ s.t. } \mathbb{E}\,\|Z\|^2 \leq C'.$$

Reintroducing a Lagrangian multiplier recovers our reguralized $\mathbb{E}\,\|Z\|^2$ objective:

$$\min \mathrm{H}[Y\,|\,Z] + \gamma\mathbb{E}\,\|Z\|^2.$$

## F.3 "Deep Variational Information Bottleneck" and $\mathbb{E}\,\|Z\|^2$

Alemi et al. (2016) model $p_\theta(z\,|\,x)$ explicitly as multivariate Gaussian with parameterized mean and parameterized diagonal covariance in their encoder and regularize it to become close to $\mathcal{N}(0, I_k)$ by

minimizing the Kullback-Leibler divergence $D_{KL}(p_\theta(z \mid x) \parallel \mathcal{N}(0, I_k))$ alongside the cross-entropy:

$$\min H_\theta[Y \mid Z] + \gamma\, D_{KL}(p(z \mid x) \parallel r(z)),$$

as detailed in section C.2.

We can expand the regularization term to

$$D_{KL}(p(z \mid x) \parallel \mathcal{N}(0, I_k))$$
$$= \mathbb{E}_{\hat{p}(x)}\, \mathbb{E}_{p(z \mid x)}\, h\left((2\pi)^{-\frac{k}{2}} e^{-\frac{1}{2}\|Z\|^2}\right) - H[Z \mid X]$$
$$= \mathbb{E}_{p(z)}\left[\frac{k}{2}\ln(2\pi) + \frac{1}{2}\|Z\|^2\right] - H[Z \mid X].$$

After dropping constant terms (as they don't matter for optimization purposes), we obtain

$$= \frac{1}{2}\mathbb{E}\,\|Z\|^2 - H[Z \mid X].$$

When we inject zero-entropy noise into the latent Z, we have $H[Z \mid X] \geq 0$ and thus $\mathbb{E}\,\|Z\|^2 - H[Z \mid X] \leq \mathbb{E}\,\|Z\|^2$. Thus, the $\mathbb{E}\,\|Z\|^2$ regularizer also upper-bounds DVIB's regularizer in this case.

In particular, we have equality when we use a deterministic encoder. When we inject zero-entropy noise and use a deterministic encoder, we are optimizing the DVIB objective function when we use the $\mathbb{E}\,\|Z\|^2$ regularizer. In other words, in this particular case, we could reinterpret "$\min H_\theta[Y \mid Z] + \gamma\,\mathbb{E}\,\|Z\|^2$" as optimizing the DVIB objective from Alemi et al. (2016) if they were using a constant covariance instead of parameterizing it in their encoder. This does not hold for stochastic encoders.

We empirically compare DVIB and the surrogate objectives from section 3.3 in section G.5. In the corresponding plot in figure G.15, we can indeed note that $\mathbb{E}\,\|Z\|^2$ and DVIB are separated by a factor of 2 in the Lagrange multiplier.

### F.4 DETAILED COMPARISON TO CEB, VCEB & DVIB

In Fisher (2019), the introduced CEB objective "$\min I[X; Z \mid Y] - \gamma I[Y; Z]$" is rewritten to "$\min \gamma H[Y \mid Z] + H[Z \mid Y] - H[Z \mid X]$" similar to the IB objective in proposition 1 in section 3.1. However, these atomic quantities are not separately examined in detail.

Both Alemi et al. (2016) and Fischer (2020) focus on the application of variational approximations to these quantities. Using a slight abuse of notation to denote all variational approximations, we can write the VCEB objective[13] (Fischer, 2020) and the DVIB objective (Alemi et al., 2016) more concisely as

$$\text{VCEB} \equiv \min_\theta H_\theta[Y \mid Z] + \beta'(H_\theta[Z \mid Y] - H_\theta[Z \mid X]),$$

$$\text{DVIB} \equiv \min_\theta H_\theta[Y \mid Z] + \beta''(H_\theta[Z] - H_\theta[Z \mid X]).$$

DVIB does not specify how to choose stochastic encoders and picks the variational marginal $q(z)$ to be a unit Gaussian. We relate how this choice of marginal relates to the $\mathbb{E}\,\|Z\|^2$ surrogate objective in section F.3. Alemi et al. (2016) use VAE-like encoders that output mean and standard deviation for latents that are then sampled from a multivariate Gaussian distribution with diagonal covariance in their experiments. They run experiments on MNIST and on features extracted from the penultimate layer of pretrained models on ImageNet.

While VCEB as introduced in Fisher (2019) is agnostic to the choice of stochastic encoder, Fischer (2020) mention that stochastic encoders can be similar to encoders and decoders in VAEs (Kingma and Welling, 2013) or like in DVIB mentioned above. Both VAEs and DVIB explicitly parameterize the distribution of the latent to sample from it before passing samples to the decoder.

Fischer and Alemi (2020) use an existing classifier architecture to output means for a Gaussian distribution with unit diagonal covariance. They further parameterize the variational approximation for the Reverse Decoder Uncertainty $q(y \mid z)$ with one Gaussian of fixed variance per class and learn this reverse decoder during training as well. Fischer and Alemi (2020) report results on CIFAR-10 and ImageNet that show good robustness against adversarial attacks without adversarial training, similar to the results in this paper.

---

[13]We will not examine the original objective without Lagrange multipliers from Fisher (2019) here.

This specific (and not motivated) instantiation of the VCEB objective in Fischer and Alemi (2020) is similar to the $\log \text{Var}[Z\,|\,Y]$ surrogate objective introduced in section 3.3 with a deterministic encoder and zero-entropy noise injection. However, the latter uses minibatch statistics instead of learning a reverse decoder, trading variational tightness for ease of computation and optimization.

Compared to this prior literature, this paper examines the usage of implicit stochastic encoders (for example when using dropout) and presents three different simple surrogate objectives together with a principled motivation for zero-entropy noise injection, which has a dual use in enforcing meaningful compression and in simplifying the estimation of information quantities. Moreover, multi-sample approaches are examined to differentiate between Decoder Cross-Entropy and Prediction Cross-Entropy. In particular, implicit stochastic encoders together with zero-entropy noise and simple surrogates make it easier to use IB objectives in practice compared to using explicitly parameterized stochastic encoders and variational approaches.

### F.5   AN INFORMATION-THEORETIC APPROACH TO VAEs

While Alemi et al. (2016) draw a general connection to $\beta$-VAEs (Higgins et al., 2016), we can use the insights from this paper to derive a simple VAE objective. Taking the view that VAEs learn latent representations that compress input samples, we can approach them as entropy estimators. Using $H[X] + H[Z\,|\,X] = H[X\,|\,Z] + H[Z]$, we obtain the ELBO

$$H[X] = H[X\,|\,Z] + H[Z] - H[Z\,|\,X] \overset{(1)}{\le} H_\theta[X\,|\,Z] + H[Z] - H[Z\,|\,X] \overset{(2)}{\le} H_\theta[X\,|\,Z] + H[Z]. \quad (40)$$

We can also put eq. (40) into words: we want to find latent representations such that the reconstruction cross-entropy $H[X\,|\,Z]$ and the latent entropy $H[Z]$, which tell us about the length encoding an input sample, become minimal and approach the true entropy as average optimal encoding length of the dataset distribution.

The first inequality (1) stems from introducing a cross-entropy approximation $H_\theta[X\,|\,Z]$ for the conditional entropy $H[X\,|\,Z]$. The second inequality (2) stems from injection of zero-entropy noise with a stochastic encoder. For a deterministic encoder, we would have equality. We also note that (1) is the DVIB objective for a VAE with $\beta = 1$, and (2) is the DIB objective for a VAE.

Finally, we can use one of the surrogates introduced in section 3.3 to upper bound $H[Z]$. For optimization purposes, we can substitute the L2 activation regularizer $\mathbb{E}\,\|Z\|^2$ from proposition 4 and obtain as objective

$$\min_\theta H_\theta[X\,|\,Z] + \mathbb{E}\,\|Z\|^2.$$

It turns out that this objective is examined amongst others in the recently published Ghosh et al. (2019) as a *CV-VAE*, which uses a deterministic encoder and noise injection with constant variance. The paper derives this objective by noticing that the explicit parameterizations that are commonly used for VAEs are cumbersome, and the actual latent distribution does often not necessarily match the induced distribution (commonly a unit Gaussian) which causes sampling to generate out-of-distribution data. It fits a separate density estimator on $p(z)$ after training for sampling. The paper goes on to then examine other methods of regularization, but also provides experimental results on CV-VAE, which are in line with VAEs and WAEs. The derivation and motivation in the paper is different and makes no use of information-theoretic principles. Our short derivation above shows the power of using the insights from section 3.2 and 3.3 for applications outside of supervised learning.

### F.6   SOFT CLUSTERING BY ENTROPY MINIMIZATION WITH GAUSSIAN NOISE

Consider the problem of minimizing $H[Z\,|\,Y]$ and $H[Y\,|\,Z]$, in the setting where $Z = f_\theta(X) + \epsilon \sim \mathcal{N}(0, \sigma^2)$—i.e. the embedding $Z$ is obtained by adding Gaussian noise to a deterministic function of the input. Let the training set be enumerated $x_1, \ldots, x_n$, with $\mu_i = f_\theta(x_i)$. Then the distribution of $Z$ is given by a mixture of Gaussians with the following density, where $d(x, \mu_i) := \|x - \mu_i\|/\sigma^2$.

$$p(z) \propto \frac{1}{n} \sum_{i=1}^{n} \exp(-d(z, \mu_i))$$

Assuming that each $x_i$ has a deterministic label $y_i$, we then find that the conditional distributions $p(y \mid z)$ and $p(z \mid y)$ are given as follows:

$$p(z \mid y) \propto \frac{1}{n_y} \sum_{i:y_i=y} \exp(-d(z, \mu_i))$$

$$p(y \mid z) = \sum_{i:y_i=y} p(\mu_i \mid z) = \sum_{i:y_i=y} \frac{p(z \mid \mu_i)\, p(\mu_i)}{p(z)}$$

$$= \frac{\sum_{i:y_i=y} p(z \mid \mu_i)}{\sum_{k=1}^n p(z \mid \mu_k)} = \frac{\sum_{i:y_i=y} \exp(-d(z, \mu_i))}{\sum_{k=1}^n \exp(-d(z, \mu_k))},$$

where $n_y$ is the number of $x_i$ with class $y_i = y$. Thus, the conditional $Z|Y$ can be interpreted as a mixture of Gaussians and $Y|Z$ as a Softmax marginal with respect to the distances between $Z$ and the mean embeddings. We observe that $H[Z \mid Y]$ is lower-bounded by the entropy of the random noise added to the embeddings:

$$H[Z \mid Y] \geq H[f_\theta(X) + \epsilon \mid Y] \geq H[\epsilon]$$

with equality when the distribution of $f_\theta(X)|Y$ is deterministic – that is $f_\theta$ is constant for each equivalence class.

Further, the entropy $H[Y \mid Z]$ is minimized when $H[Z]$ is large compared to $H[Z \mid Y]$ as we have the decomposition

$$H[Y \mid Z] = H[Z \mid Y] - H[Z] + H[Y].$$

In particular, when $f_\theta$ is constant over equivalence classes of the input, then $H[Y \mid Z]$ is minimized when the entropy $H[f_\theta(X) + \epsilon]$ is large – i.e. the values of $f_\theta(x_i)$ for each equivalence class are distant from each other and there is minimal overlap between the clusters. Therefore, the optima of the information bottleneck objective under Gaussian noise share similar properties to the optima of geometric clustering of the inputs according to their output class.

To gain a better understanding of local optimization behavior, we decompose the objective terms as follows:

$$\begin{aligned}
H[Z \mid Y] &= \mathbb{E}_{\hat{p}(y)}\, H(p(z \mid y) \| p(z \mid y)) \\
&= \mathbb{E}_{\hat{p}(x,y)}\, H(p(z \mid x) \| p(z \mid y)) \\
&= \mathbb{E}_{\hat{p}(x,y)}\, D_{KL}(p(z \mid x) \| p(z \mid y)) + H[Z \mid x] \\
&= \mathbb{E}_{\hat{p}(x,y)}\, D_{KL}(p(z \mid x) \| p(z \mid y)) \\
&\quad + \underbrace{H[Z \mid X]}_{=const}.
\end{aligned}$$

To examine how the mean embedding $\mu_k$ of a single datapoint $x_k$ affects this entropy term, we look at the derivative of this expression with respect to $\mu_k = f_\theta(x_k)$. We obtain:

$$\begin{aligned}
\frac{d}{d\mu_k} H[Z \mid Y] &= \frac{d}{d\mu_k} H[Z \mid y_k] \\
&= \frac{d}{d\mu_k} \mathbb{E}_{p(x|y_k)}\, D_{KL}(p(z \mid x) \| p(z \mid y)) \\
&= \sum_{i \neq i: y_i = y_k} \frac{1}{n_{y_k}} \frac{d}{d\mu_k} D_{KL}(p(z \mid x_i) \| p(z \mid y_k)) \\
&\quad + \frac{1}{n_{y_k}} \frac{d}{d\mu_k} D_{KL}(p(z \mid x_k) \| p(z \mid y_k)).
\end{aligned}$$

While these derivatives do not have a simple analytic form, we can use known properties of the KL divergence to develop an intuition on how the gradient will behave. We observe that in the left-hand sum $\mu_k$ only affects the distribution of $Z|Y$ (that is we are differentiating a sum of terms that look like a reverse KL), whereas it has greater influence on $p(z \mid x_k)$ in the right-hand term, and so its gradient will more closely resemble that of the forward KL. The left-hand-side term will therefore push $\mu_k$ towards the centroid of the means of inputs mapping to $y$, whereas the right-hand side term is mode-seeking.

### F.7 A NOTE ON DIFFERENTIAL AND DISCRETE ENTROPIES

The mutual information between two random variables can be defined in terms of the KL divergence between the product of their marginals and their joint distribution. However, the KL divergence is only well-defined when the Radon-Nikodym derivative of the density of the joint with respect to the product exists. Mixing continuous and discrete distributions—and thus differential and continuous entropies—can violate this requirement, and so lead to negative values of the "mutual information". This is particularly worrying in the setting of training stochastic neural networks, as we often assume that an stochastic embedding is generated as a deterministic transformation of an input from a finite dataset to which a continuous perturbation is added. We provide an examples where naive computation without ensuring that the product and joint distributions of the two random variables have a well-defined Radon-Nikodym derivative yields negative mutual information.

Let $X \sim U([0, 0.1])$, $Z = X + R$ with $R \sim U(\{0, 1\})$. Then
$$I[X; Z] = H[X] = \log \tfrac{1}{10} \leq 0.$$
Generally, given $X$ as above and an invertible function $f$ such that $Z = f(X)$, $I[X; Z] = H[X]$ and can thus be negative. In a way, these cases can be reduced to (degenerate) expressions of the form $I[X; X] = H[X]$.

We can avoid these cases by adding independent continuous noise.

These examples show that not adding noise can lead to unexpected results. While they still yield finite quantities that bear a relation to the entropies of the random variables, they violate some of the core assumptions we have such that mutual information is always positive.

## G EXPERIMENT DETAILS

### G.1 DNN ARCHITECTURES AND HYPERPARAMETERS

For our experiments, we use PyTorch (Paszke et al., 2019) and the Adam optimizer (Kingma and Ba, 2014). In general, we use an initial learning rate of $0.5 \times 10^{-3}$ and multiply the learning rate by $\sqrt{0.1}$ whenever the loss plateaus for more than 10 epochs for CIFAR-10. For MNIST and Permutation MNIST, we use an initial learning rate of $10^{-4}$ and multiply the learning rate by 0.8 whenever the loss plateaus for more than 3 epochs.

Sadly, we deviate from this in the following experiments: when optimizing the decoder uncertainty for *categorical Z* for CIFAR-10, we used 5 epochs patience for the decoder uncertainty objective and a initial learning rate of $10^{-4}$. We do not expect this difference to affect the qualitative results mentioned in section E when comparing to other objectives. We also only used 5 epochs patience when comparing the two cross-entropies on CIFAR-10 in section 3.2. As this was used for both sets of experiments, it does not matter.

We train the experiments for creating the information plane plots for 150 epochs. The toy experiment (figure 3) is trained for 20 epochs. All other experiments train for 100 epochs.

We use a batchsize of 128 for most experiments. We use a batchsize of 32 for comparing the cross-entropies for CIFAR-10 (where we take 8 dropout samples each), and a batchsize of 16 for MNIST (where we take 64 dropout samples each).

For MNIST, we use a standard dropout CNN, following `https://github.com/pytorch/examples/blob/master/mnist/main.py`. For Permutation MNIST, we use a fully-connected model (for experiments with categorical Z in section E): $784 \times 1024 \times 1024 \times C$. For CIFAR-10, we use a regular deterministic ResNet18 model (He et al., 2016a) for the experiments in section E. (As the model outputs a categorical distribution it becomes stochastic through that and we don't need stochasticity in the weights.) For the other experiments as well as the Imagenette experiments, we use a ResNet18v2 (He et al., 2016b). When we need a stochastic model for CIFAR-10 (for *continuous Z*), we add DropConnect (Wan et al., 2013b) with rate 0.1 to all but the first convolutional layers and dropout with rate 0.1 before the final fully-connected layer. Because of memory issues, we reuse the dropout masks within one batch. The model trains to 94% accuracy on CIFAR-10.

For CIFAR-10, we always remove the maximum pooling layer and change the first convolutional layer to have kernel size 3 with stride 1 and padding 1. We also use dataset augmentation during training, but not during evaluation on the training set and test set for purposes of computing metrics. We crop randomly after the padding the training images by 4 pixels in every direction and randomly flip images horizontally.

We generally sample 30 values of $\gamma$ for the information plane plots from the specified ranges, using a log scale. For the ablation studies mentioned below, we sample 10 values of $\gamma$ each. We always sample $\gamma = 0$ separately and run a trial with it.

Baselines were tuned by hand (without regularization) using grad-student descent and small grid searches.

### G.2 CLUSTER SETUP & USED RESOURCES

We make use of a local SLURM cluster (Jette et al., 2002). We run our experiments on GPUs (Geforce RTX 2080 Ti). We estimate reproducing all results would take 94 GPU days.

### G.3 COMPARISON OF THE SURROGATE OBJECTIVES

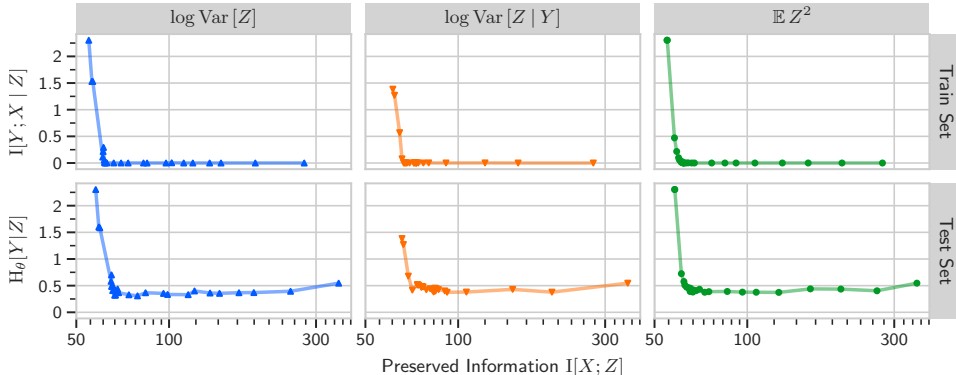

Figure G.1: *Information Plane Plot of the latent* Z *similar to* Tishby and Zaslavsky (2015) *but using a* ResNet18 *model on* CIFAR-10 *using the different regularizes from section 3.3 (*without dropout, but with zero-entropy noise*). The dots are colored by $\gamma$. See section 4 for more details.*

As can be seen in figure G.2, the different surrogate regularizers have very similar effects on H[Z] and H[Z | Y]. Regularizing with $\mathbb{E} \|Z\|^2$ shows a stronger initial regularization effect, but is difficult to compare quantitatively as its hyperparameter does not map to an equivalent $\beta$, unlike regularizing using entropy estimates. Overall, we find log Var[Z] to provide stable training trajectories (and expected visualizations) while also having a more meaningful hyperparameter than $\mathbb{E} \|Z\|^2$, though $\mathbb{E} \|Z\|^2$ is trivial to implement and communicate[14]. log Var[Z | Y] performs worse which we hypothesize is due to the increased variance (given equal batch sizes) from conditioning on Y. It further does not minimize the Preserved Information I[X; Z] as strongly as the other regularizers.

### G.3.1 MEASUREMENT OF INFORMATION QUANTITIES

Measuring information quantities can be challenging. As mentionend in the introduction, there are many complex ways of measuring entropies and mutual information terms. We can side-step the issue by making use of the bounds we have established and the zero-entropy noise we are injecting, and design experiments around that.

First, to estimate the Preserved Information I[X; Z], we note that when we use a deterministic model as encoder and only inject zero-entropy noise, we have H[Z | X] = 0 and I[X; Z] = I[X; Z] + H[Z | X] = H[Z]. We use the entropy estimator from Kraskov et al. (2004, equation (20)) to estimate the Encoding Entropy H[Z] and thus I[X; Z].

---

[14]Which is the reason why we showcase it in figure 1 and in equation (1).

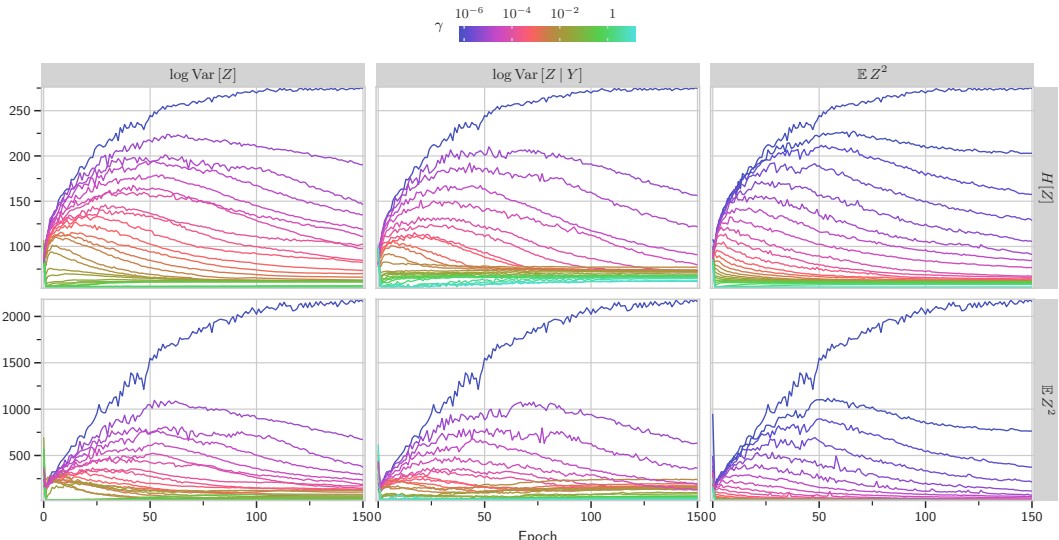

Figure G.2: *Entropy estimates while training with different γ and with different surrogate regularizers on CIFAR-10 with a ResNet18 model.* Entropies are estimated on training data based on Kraskov et al. (2004). Qualitatively all three regularizers push H[Z] and H[Z | Y] down. H[Z | Y] is not shown here because it always stays very close to H[Z]. $\mathbb{E} \|Z\|^2$ tends to regularize entropies more strongly for small γ. See section 4 for more details.

To estimate the Residual Information I[X; Y | Z], we similarly note that I[X; Y | Z] = I[X; Y | Z] + H[Y | X] = H[Y | Z]. Instead of estimating the entropy using Kraskov et al. (2004), we can use the Decoder Cross-Entropy $H_\theta$[Y | Z] which provides a tighter bound as long as we also minimize $H_\theta$[Y | Z] as part of the training objective.

When we use stochastic models as encoder, we cannot easily compute I[X; Z] anymore. In the ablation study in the next section, we thus change the X axis accordingly.

Similarly, when we look at the trajectories on the test set instead of the training set, for example in figure G.4, we change the Y axis to signify the Decoder Uncertainty $H_\theta$[Y | Z]. It is still an upper-bound, but we do not minimize it directly anymore.

For the plots in figure 6, we retrained the decoder on the test set to obtain a tighter bound on H[Y | Z] (while keeping the encoder fixed). We then sampled the latent using the test set to estimate the trajectories. We only did this for the CIFAR-10 model without dropout. For our ablations, we did not retrain the decoder and thus only present plots on the test and training set, respectively.

At this point, it is important to recall that the Decoder Uncertainty is also the negative log-likelihood (when training with a single dropout sample), which provides a different perspective on the plots. It makes it clear that we can see how much a model overfits by comparing the best and final epochs of a trajectory in the plot (marked by a circle and a square, respectively).

### G.3.2    ABLATION STUDY

We perform an ablation study to determine whether injecting noise is necessary. Furthermore, we investigate the more interesting case of using a stochastic model as encoder, and if we can use a stochastic model without injecting zero-entropy noise.

We also investigate whether log Var[Z | Y] performs better when we increase batchsize as we hypothesized that a batchsize of 128 does not suffice as it leaves only ≈ 13 samples per class to approximate H[Z | Y]).

Figure G.3 shows a larger version of figure 6 for all three regularizers and also training trajectories on the test set. As described in the previous section, this allows us to validate that the regularizers prevent overfitting on the training set: with increasing γ, the model overfits less.

Figure G.6 and figure G.5 shows that injecting noise is necessary independently of whether we use dropout or not. Regularizing with $\mathbb{E} \|Z\|^2$ still has a very weak effect. We hypothesize that, similar to

the toy experiment depicted in figure 3, floating-point precision issues might provide a natural noise source eventually. This would change the effectiveness of $\gamma$ and might require much higher values to observe similar regularization effects as when we do inject zero-entropy noise.

Figure G.4 shows trajectories for a stochastic encoder (as described above with DropConnect/dropout rate 0.1). It overfits less than a deterministic one.

Figure G.7 shows the effects of using higher dropout rates (using DropConnect/dropout rates of 0.3/0.5). It overfits less than model with DropConnect/dropout rates of 0.1/0.1.

The plots in figure G.10 show the effects of different $\gamma$ with different regularizers more clearly. On both training and test set, one can clearly see the effects of regularization.

Overall, $\log \text{Var}[Z \,|\, Y]$ performs worse as a regularizer. In figure G.8, we compare the effect of doubling batchsize. Indeed, $\log \text{Var}[Z \,|\, Y]$ performs better with higher batchsize and looks closer to $\log \text{Var}[Z]$.

### G.3.3   Comparison between Decoder Cross-Entropy and Prediction Cross-Entropy

When training deterministic models or dropout models with a single sample (as one usually does), the estimators for both the Decoder Cross-Entropy $H_\theta[Y \,|\, Z]$ and the Prediction Cross-Entropy $H_\theta[Y \,|\, X]$ coincide. In section 3.2, we discuss the differences from a theoretical perspective. Here, we empirically evaluate the difference between optimizing the estimators for each of the two cross-entropy losses, for which we will draw multiple dropout samples during training and inference.

We examine models with *continuous* Z on MNIST and CIFAR-10 (Lecun et al., 1998; Krizhevsky et al., 2009). Specifically, we use a standard dropout CNN as an encoder for MNIST, and a modified ResNet18 to which we add DropConnect in each layer for CIFAR-10. We use $K = 100$ dimensions for the continuous latent Z in the last fully-connected layer, and use a linear decoder to obtain the final 10-dimensional output of class logits. For MNIST, we compute the cross-entropies using 64 dropout samples; for CIFAR-10, we use 8. For the purpose of this examination of training behavior, it is not necessary to achieve SOTA accuracy: our models obtain 99.2% accuracy on MNIST and 93.6% on CIFAR-10.

Figure G.11 shows the training error probability as well as the value of each cross-entropy loss for models trained either with the Decoder Cross-Entropy or the Prediction Cross-Entropy. The Decoder Cross-Entropy $H_\theta[Y \,|\, Z]$ outperforms Prediction Cross-Entropy $H_\theta[Y \,|\, X]$ as a training objective: the training error probability and both cross-entropies are lower when minimizing $H_\theta[Y \,|\, Z]$ compared to minimizing $H_\theta[Y \,|\, X]$. We compare only the training, rather than the test, losses of the models to isolate the effect of each loss term on training performance; we leave the prevention of overfitting to the regularization terms considered later. Recently, Dusenberry et al. (2020) also observed empirically that the Decoder Cross-Entropy $H_\theta[Y \,|\, Z]$ as an objective is both easier to optimize and provides better generalization performance.

### G.4   Differential entropies and noise

We demonstrate the importance of adding noise to continuous latents by constructing a pathological sequence of parameters which attain monotonically improving and unbounded regularized objective values ($H[Z]$) while all computing *the same function*. We use MNIST with a standard dropout CNN as encoder, with $K = 128$ continuous dimensions in Z, and a $K \times 10$ linear layer as decoder. After every training epoch, we decrease the entropy of the latent by normalizing and then scaling the latent to bound the entropy. We multiply the weights of the decoder to not change the overall function. As can be seen in figure 3, without noise, entropy can decrease freely during training without change in error rate until it is affected by floating-point issues; while when adding zero-entropy noise, the error rate starts increasing gradually and meaningfully as the entropy starts to approach zero. We conclude that entropy regularization is meaningful only when noise is added to the latent.

## G.5 Comparison between DVIB and surrogate objectives on Permutation-MNIST

Comparing DVIB and our surrogate objectives is not straightforward because DVIB uses a VAE-like model that explicitly parameterize mean and standard deviation of the latent whereas the stochastic models we focus on in section 3.2 and beyond are implicit by using dropout.

For this comparison, we use the same architecture and optimization strategy for DVIB as described in Alemi et al. (2016): the encoder is a ReLU-MLP of the form $796 - 1024 - 1024 - 2K$ with K=256 latent dimensions that outputs mean and standard deviation explicitly and separately. For the standard deviation, we use a softplus transform with a bias of $-5$. We use Polyak averaging with a decay constant of 0.999 (Polyak and Juditsky, 1992). We train the model for 200 epochs with Adam with learning rate $10^{-4}, \beta_1 = 0.5, \beta_2 = -.999$ (Kingma and Ba, 2014) and decay the learning rate by 0.97 every 2 epochs. The marginal is fixed to a unit Gaussian around the origin. We use a softmax layer as decoder. We use 12 latent samples during training and test time.

For our surrogate objectives, we use a similar ReLU-MLP of the form $796 - 1024 - 1024 - K$ with K=256 latent dimensions and dropout layers of rate 0.3 after the first and second layer. We use also 12 dropout samples during training and test time. We train for 75 epochs with Adam and learning rate $0.5 \times 10^{-4}$. We half the learning rate every time the loss does not decrease for 13 epochs.

We run 5 trials for each experiments. We were not able to reproduce the baseline of an error of 1.13% for $\beta = 10^{-3}$ from Alemi et al. (2016). We show a comparison in figure G.15. Our methods do reach an error of 1.13% overall though, so the simpler surrogate objectives perform as well good or better than DVIB.

From section F.3, we know that DVIB's $\beta$ would have to be twice the $\gamma$ frm our section 3.3. We can see this correspondence in the plot. This also implies that DVIB's $\beta$ is not related to the IB objective's $\beta$ from section 3. This makes sense as DVIB arbitrarily fixes the marginal to be a unit Gaussian.

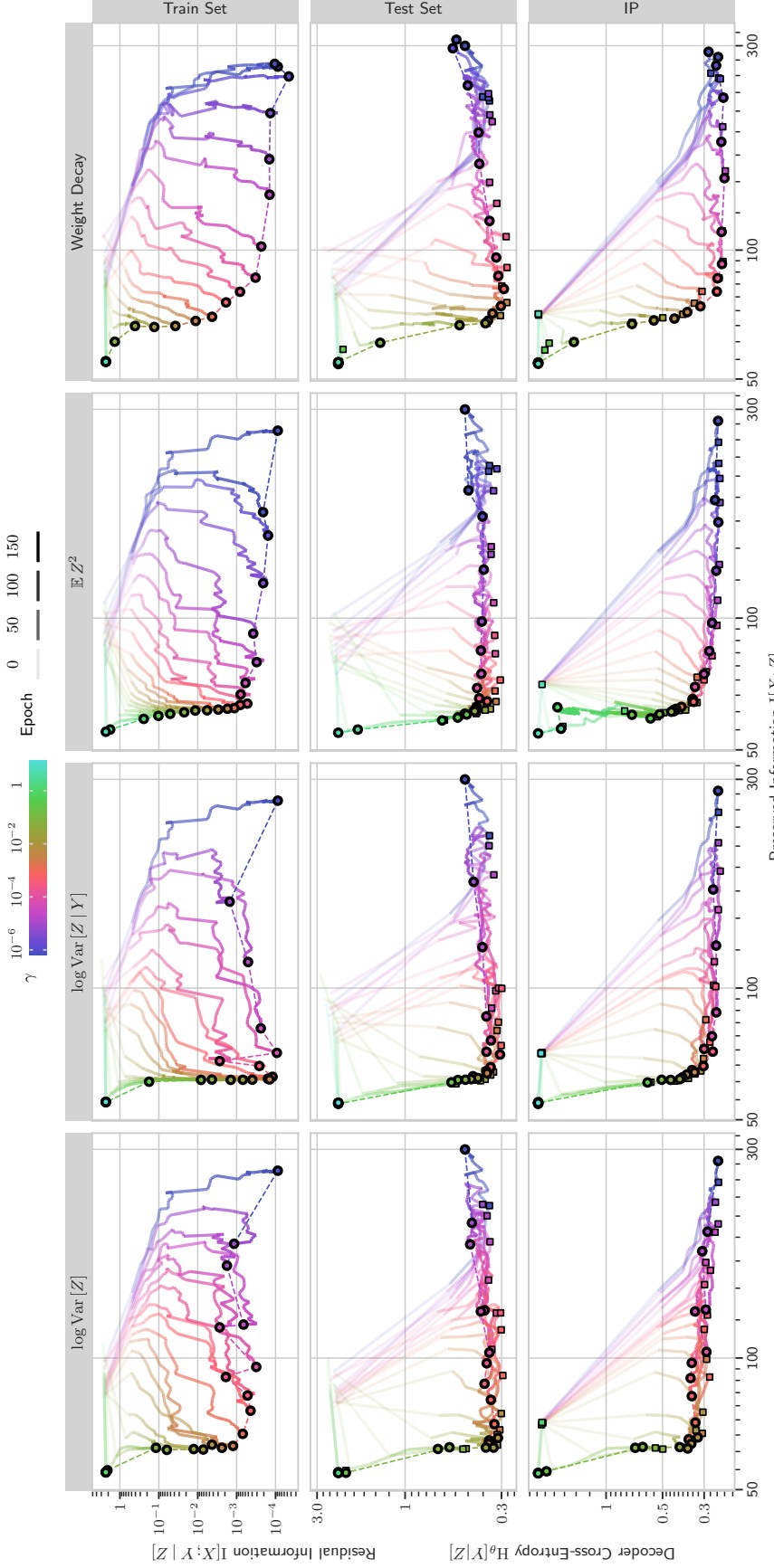

Figure G.3: *Without dropout but with zero-entropy noise: Information Plane Plot of training trajectories for ResNet18 models on CIFAR-10 and different regularizers.* The trajectories are colored by their respective $\gamma$; their transparency changes by epoch. Compression (Preserved Information $\downarrow$) trades-off with performance (Residual Information $\downarrow$). See section 4. The circle marks the final epoch of a trajectory. The square marks the best epoch (Residual Information $\downarrow\downarrow$).

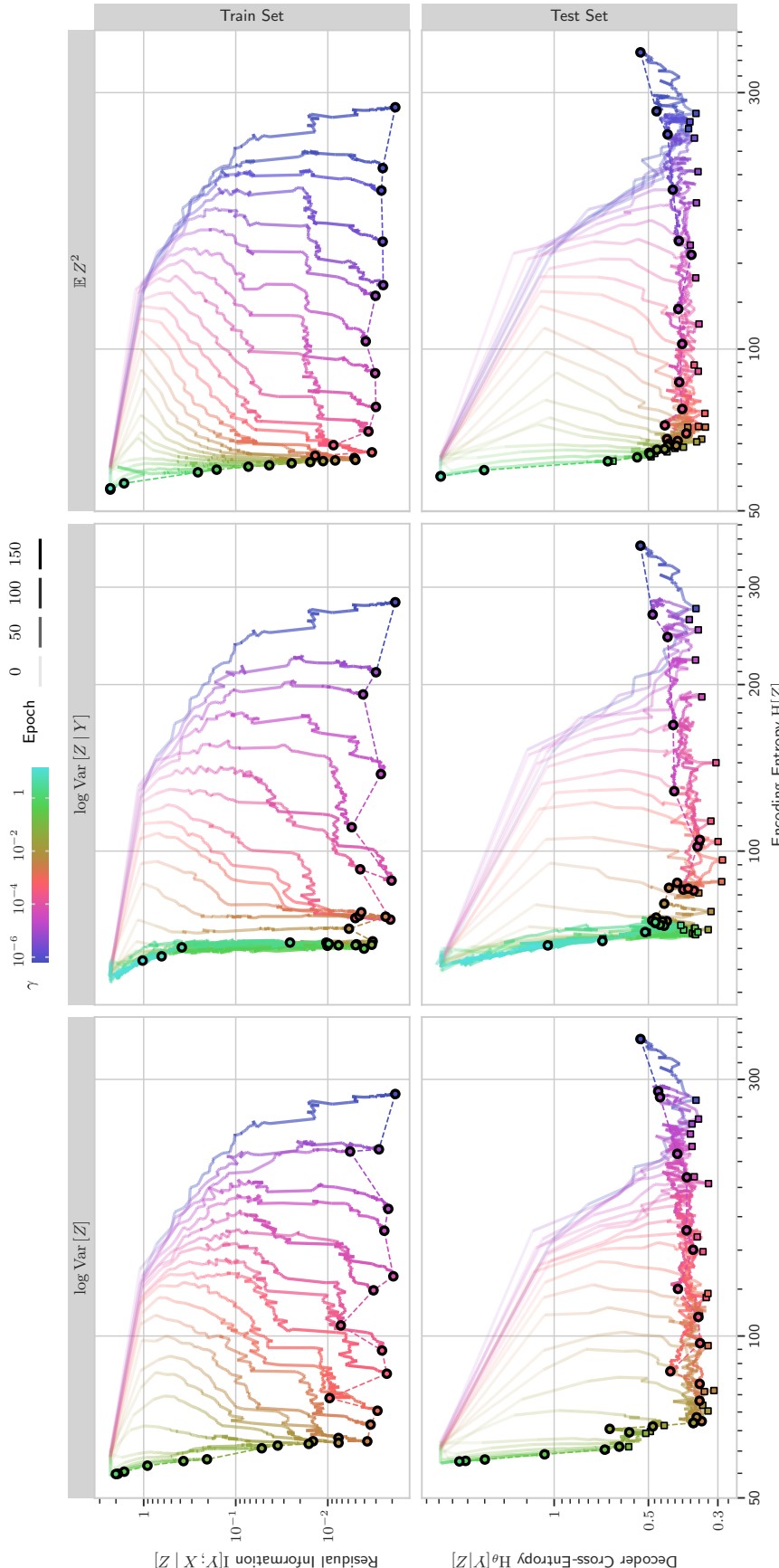

Figure G.4: With dropout and with zero-entropy noise: *Information Plane Plot of training trajectories for ResNet18 models on CIFAR-10 and different regularizers. The trajectories are colored by their respective $\gamma$; their transparency changes by epoch. Compression (Preserved Information ↓) trades-off with performance (Residual Information ↓). See section 4. The circle marks the final epoch of a trajectory. The square marks the best epoch (Residual Information ↓↓).*

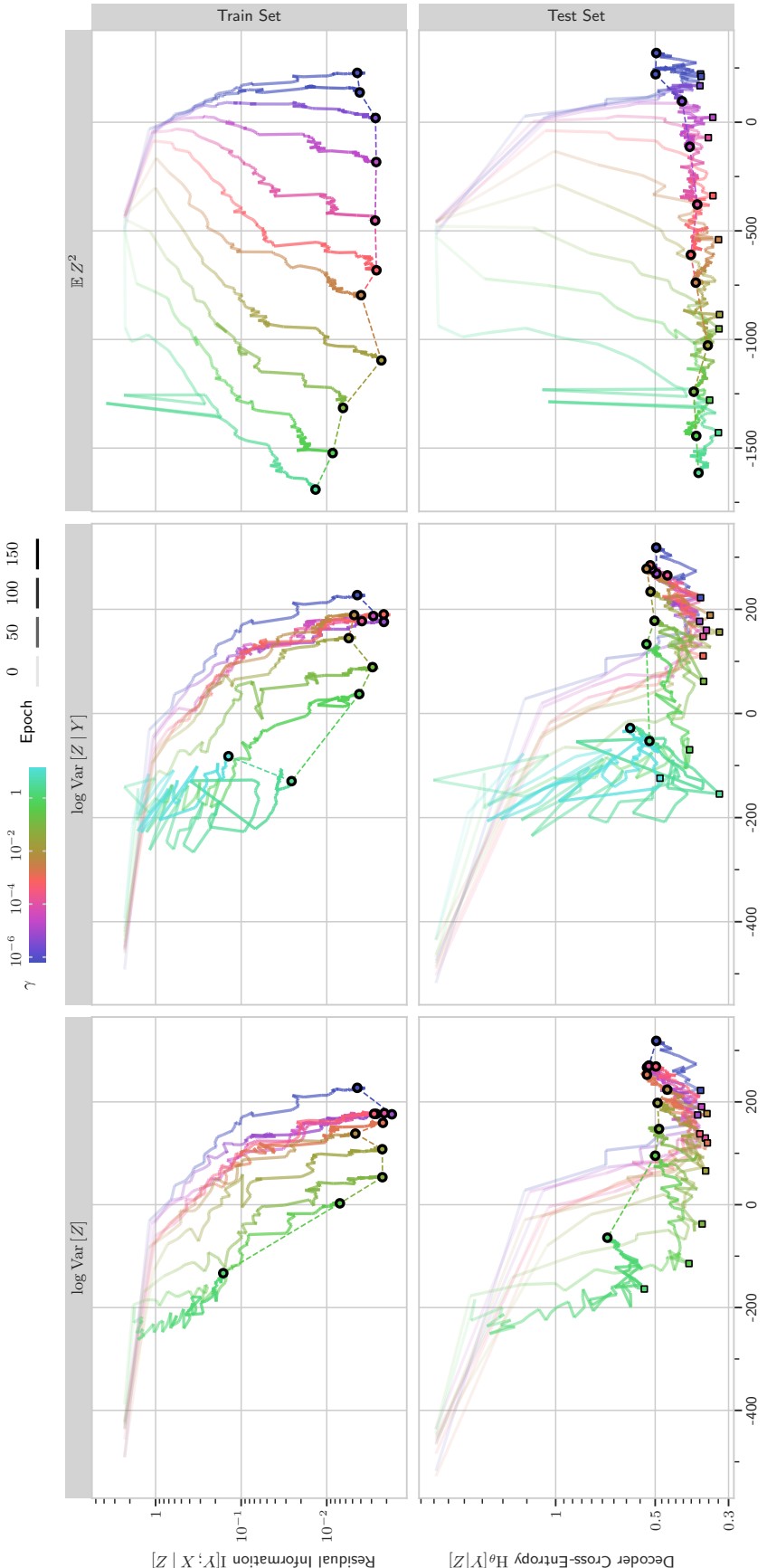

Figure G.5: With dropout but without zero-entropy noise: *Information Plane Plot of training trajectories for ResNet18 models on CIFAR-10 and different regularizers. The trajectories are colored by their respective γ; their transparency changes by epoch. Compression (Preserved Information ↓) trades-off with performance (Residual Information ↓). See section 4. The circle marks the final epoch of a trajectory. The square marks the best epoch (Residual Information ↓↓).*

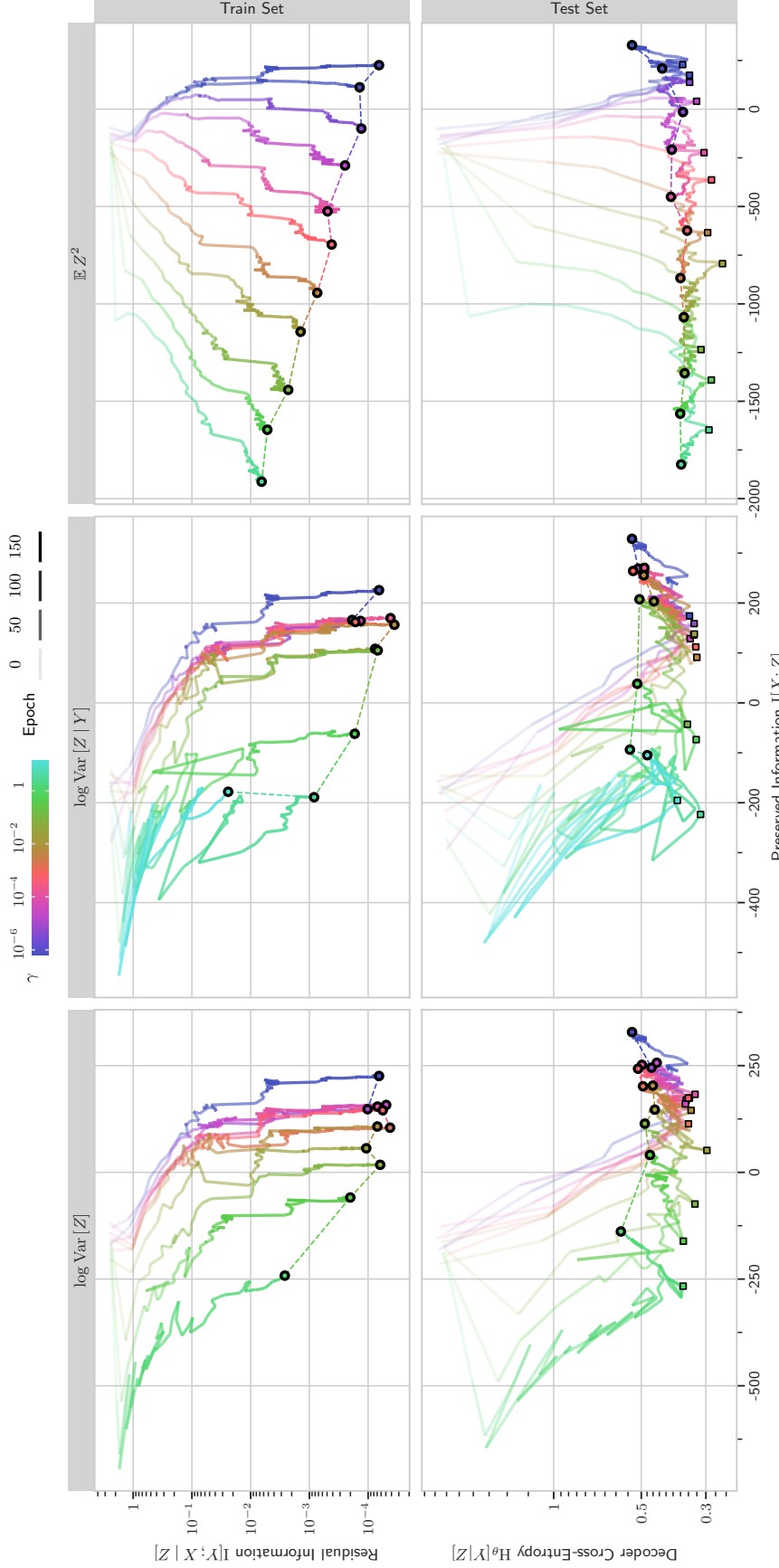

Figure G.6: Without dropout and without zero-entropy noise: *Information Plane Plot of training trajectories for ResNet18 models on CIFAR-10 and different regularizers.* The trajectories are colored by their respective γ; their transparency changes by epoch. Compression (Preserved Information ↓) trades-off with performance (Residual Information ↓). See section 4. The circle marks the final epoch of a trajectory. The square marks the best epoch (Residual Information ⇊).

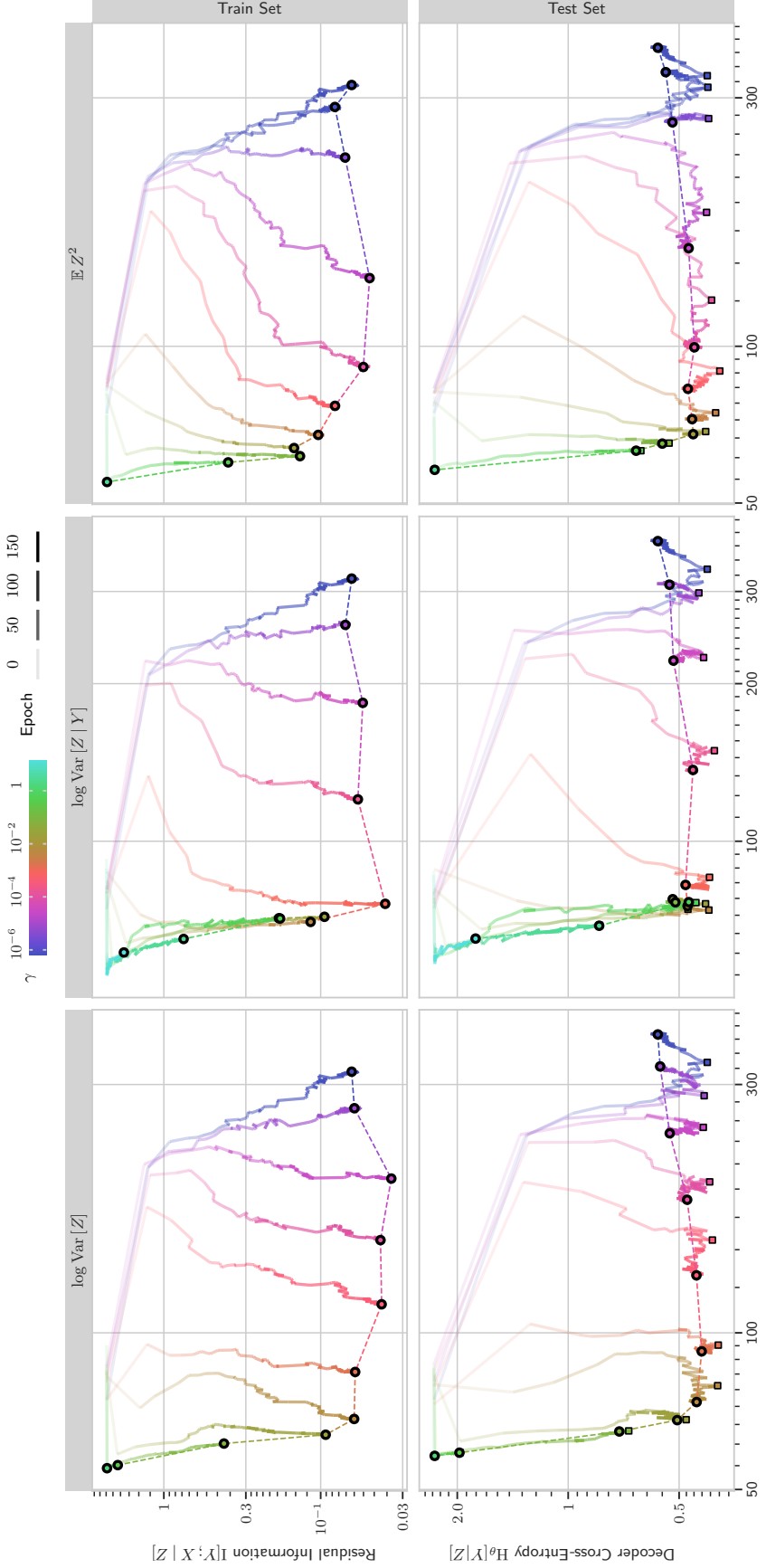

Figure G.7: With more dropout and zero-entropy noise: *Information Plane Plot of training trajectories for ResNet18 models on CIFAR-10 and* log Var[Z | Y] *regularizer with batchsizes 128 and 256.* The trajectories are colored by their respective γ; their transparency changes by epoch. Compression (Preserved Information ↓) trades-off with performance (Residual Information ↓). See section 4. The circle marks the final epoch of a trajectory. The square marks the best epoch (Residual Information ↓↓). A DropConnect rate of 0.3 and dropout rate of 0.4 were used instead of 0.1 for each.

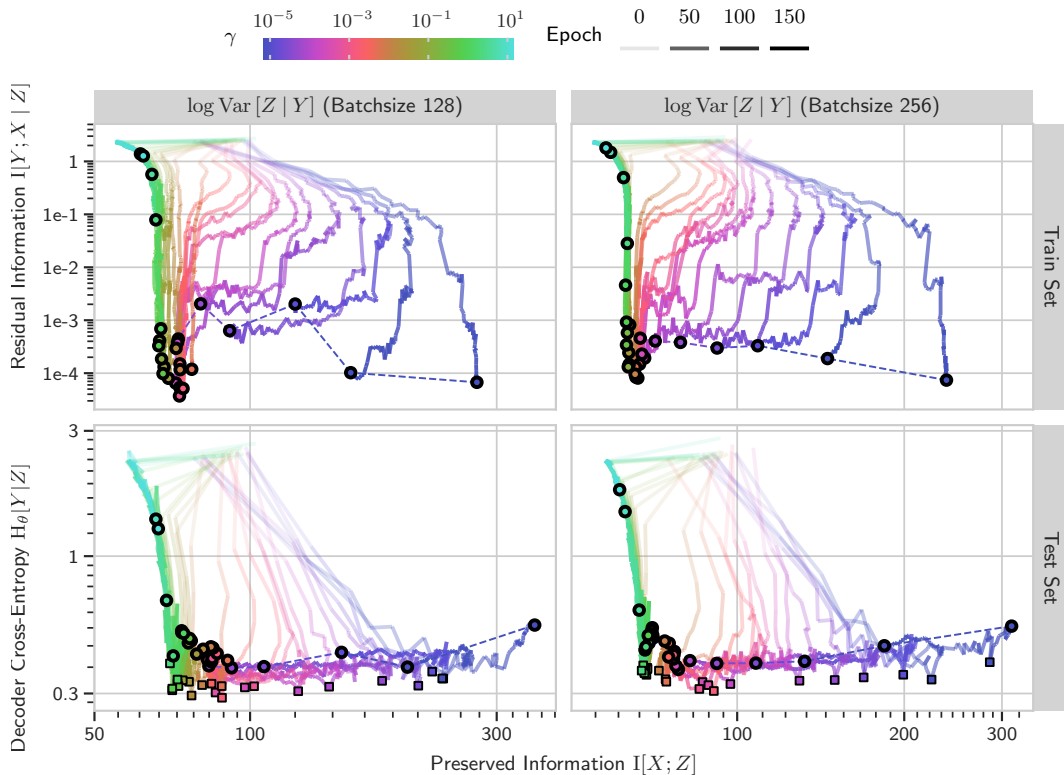

Figure G.8: Without dropout but with zero-entropy noise: *Information Plane Plot of training trajectories for ResNet18 models on CIFAR-10 and* log Var[Z | Y] *regularizer with batchsizes 128 and 256.* The trajectories are colored by their respective $\gamma$; their transparency changes by epoch. Compression (Preserved Information ↓) trades-off with performance (Residual Information ↓). See section 4. The circle marks the final epoch of a trajectory. The square marks the best epoch (Residual Information ⇊).

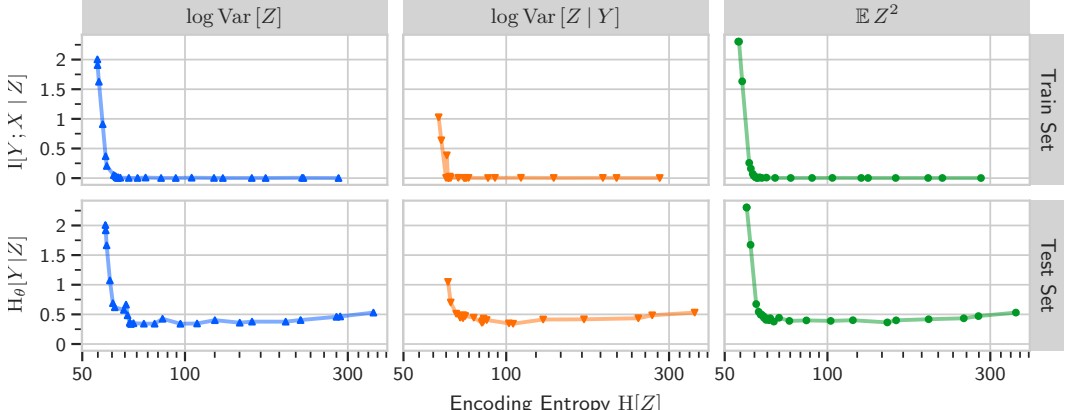

Figure G.9: *Information Plane Plot of the latent* Z *similar to* Tishby and Zaslavsky (2015) *but using a* ResNet18 *model on* CIFAR-10 *using the different regularizes from section 3.3 (with dropout and zero-entropy noise).* The dots are colored by $\gamma$. See section 4 for more details.

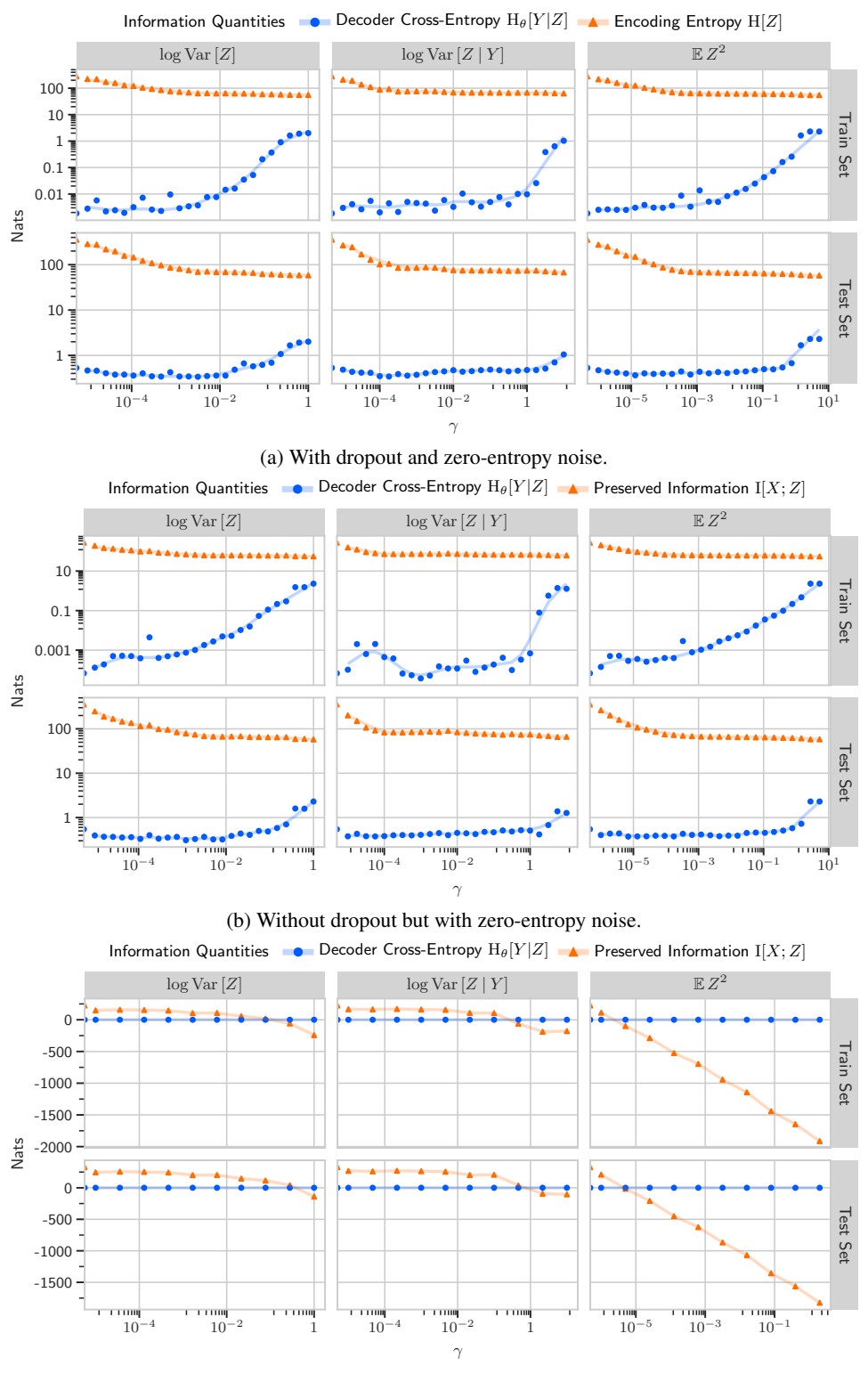

(a) With dropout and zero-entropy noise.

(b) Without dropout but with zero-entropy noise.

(c) Without dropout and without zero-entropy noise.

Figure G.10: *Information quantites for different γ at the end of training for ResNet18 models on CIFAR-10 and* log Var[Z | Y] *regularizer with batchsizes 128 and 256.* Compression (Preserved Information ↓) trades-off with performance (Residual Information ↓). See section 4.

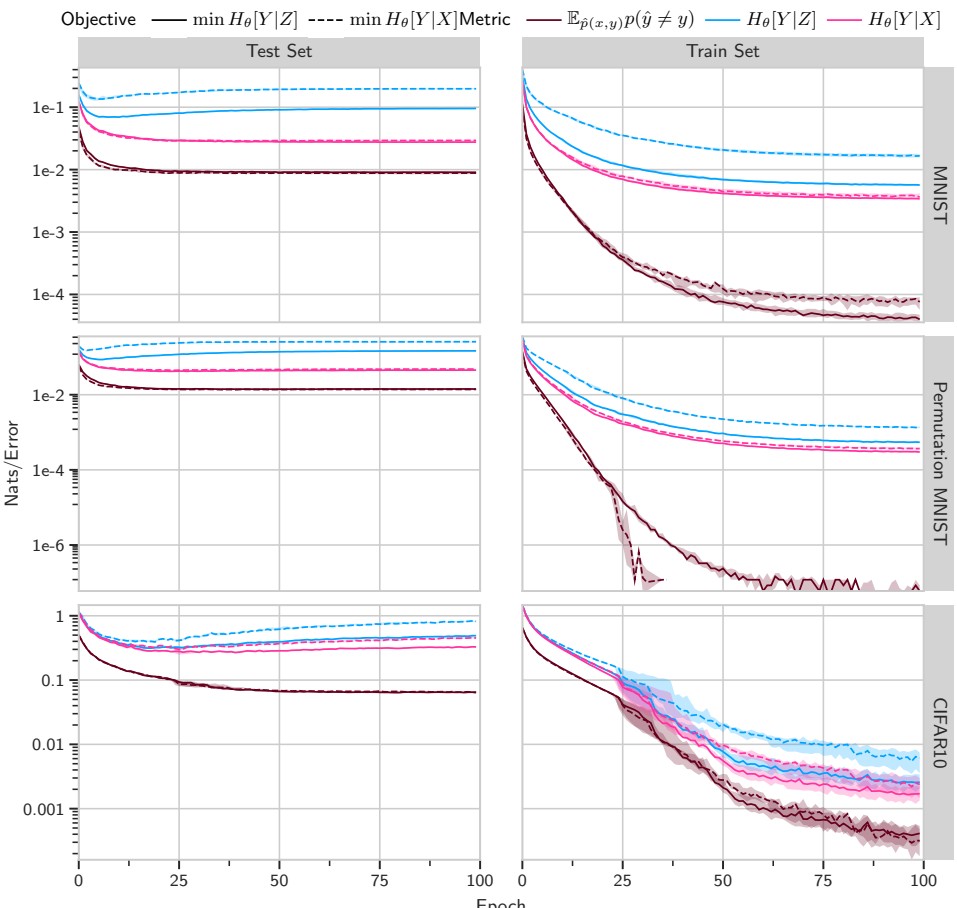

Figure G.11: *Training error probability, Decoder Cross-Entropy* $H_\theta[Y \mid Z]$ *and Prediction Cross-Entropy* $H_\theta[Y \mid X]$ *with continuous* Z. $K = 100$ *dimensions are used for Z, and we use dropout to obtain stochastic models. Minimizing* $H_\theta[Y \mid Z]$ *(solid) leads to smaller cross-entropies and lower training error probability than minimizing* $H_\theta[Y \mid X]$ *(dashed). This suggests a better data fit, which is what we desire for a loss term. We run 8 trials each and plot the median with confidence bounds (25% and 75% quartiles). See section 3.2 and G.3.3 for more details.*

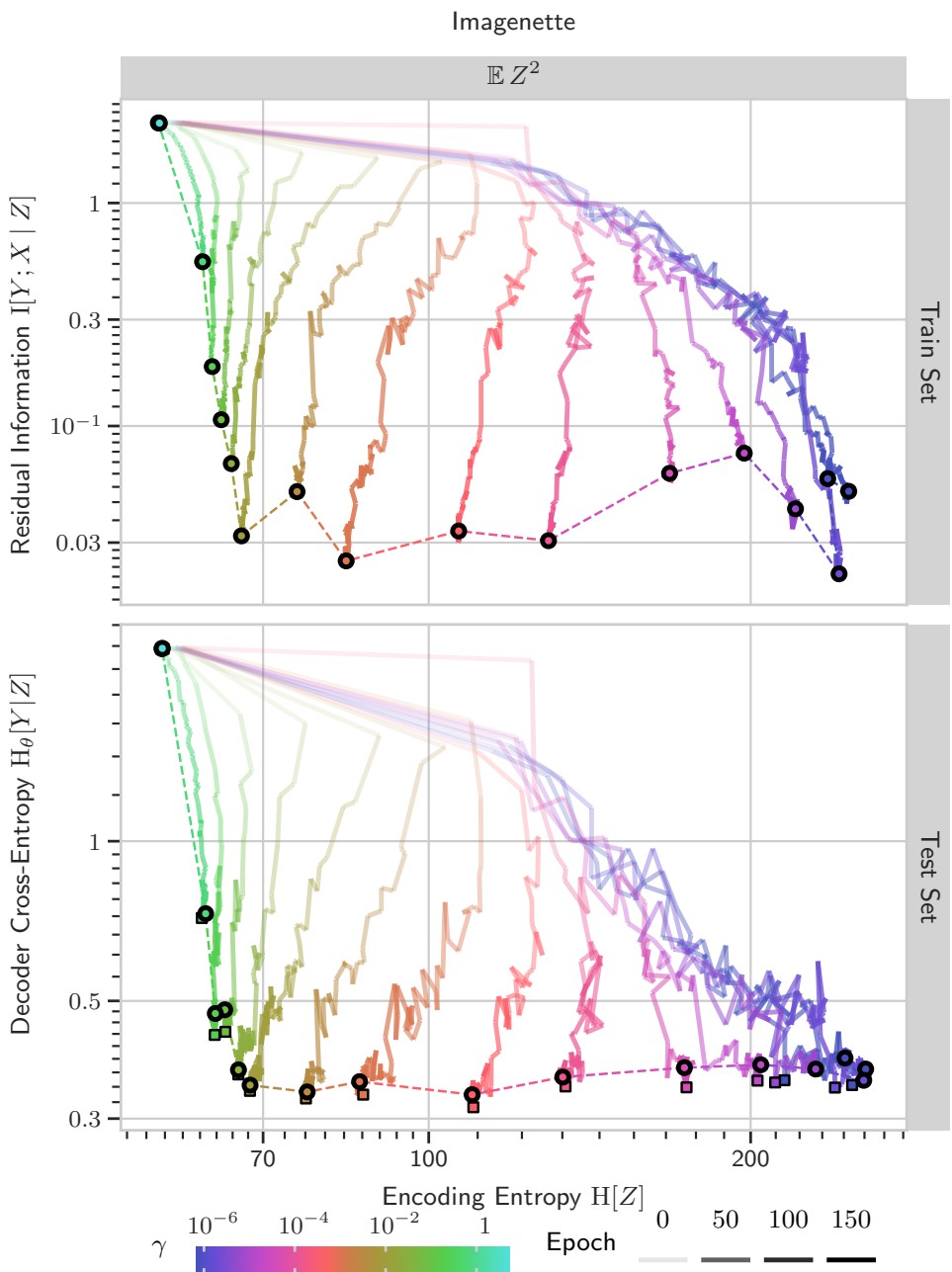

Figure G.12: *Information Plane Plot of the latent* Z *similar to* Tishby and Zaslavsky (2015) *but using a* ResNet18v2 *model on* Imagenette *using the* $\mathbb{E}\|Z\|^2$ *surrogate obejctive from section 3.3 (with dropout and zero-entropy noise). The dots are colored by* $\gamma$.

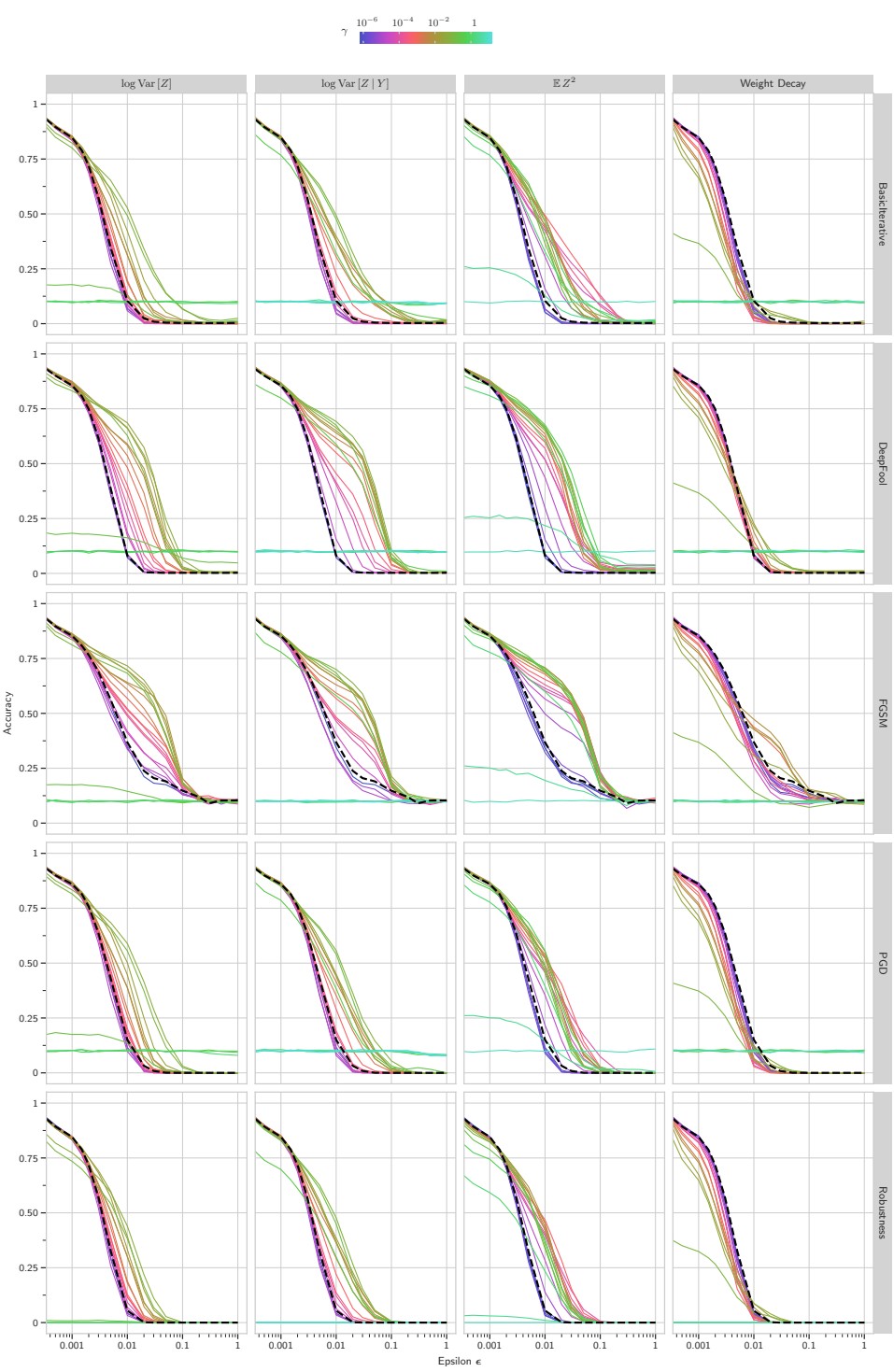

Figure G.13: *Adversarial robustness of ResNet18 models trained on CIFAR-10 with surrogate objectives in comparison to regularization with L2 weight-decay as non-IB method for different attack strengths $\epsilon$.* The robustness is evaluated using FGSM, PGD, DeepFool and BasicIterative attacks of varying $\epsilon$ values. The dashed black line represents a model trained only with cross-entropy and no noise injection. We see that models trained with the surrogate IB objective (colored by $\gamma$) see improved robustness over a model trained only to minimize the cross-entropy training objective (shown in black) while the models regularized with weight-decay actually perform worse.

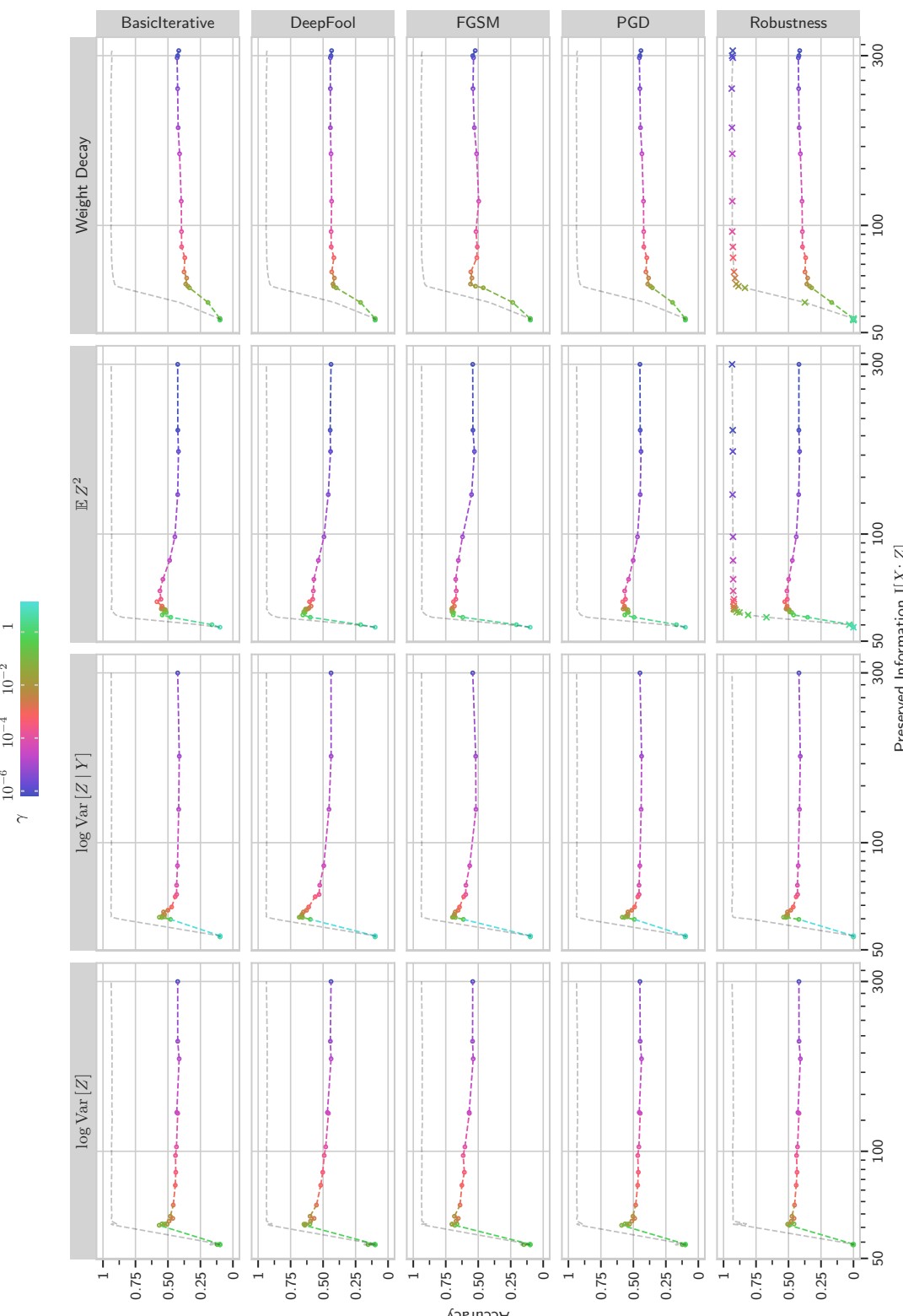

Figure G.14: *Average adversarial robustness over $\epsilon \in [0, 0.1]$ of ResNet18 models trained on CIFAR-10 with surrogate objectives in or L2 weight-decay (as non-IB method) compared to normal accuracy for different amounts of* Preserved Information. *o markers show the normal accuracy. × markers show robustness.* Preserved Information. *If the latent is compressed too much, robustness (and accuracy) are low. If the latent is not compressed enough, robustness depends on the* Preserved Information. *If the latent is compressed too much, robustness (and accuracy) are low. If the latent is not compressed enough, robustness and thus generalization suffer.*

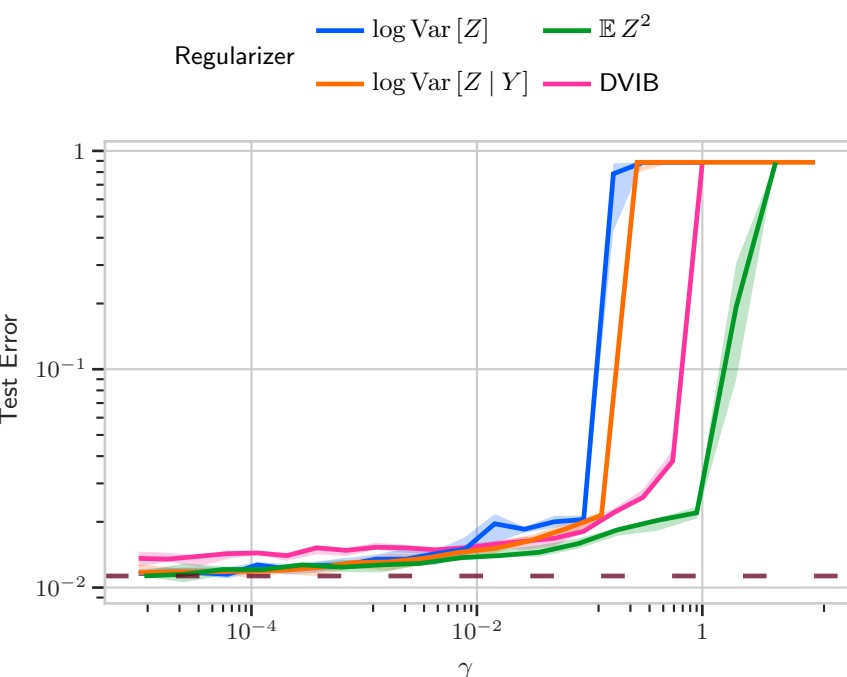

Figure G.15: *Comparison of test error for different Lagrange multiplier for DVIB and surrogate objectives from section 3.3 on Permutation-MNIST.* The purple strongly dashed line shows the test error reported for DVIB in Alemi et al. (2016). 5 trials with 95% confidence interval shown. Even though we could not reproduce the baseline reported in that paper, the simpler surrogate objective reach at least a similar test error as reported there. We also see that DVIB behaves similar to $\mathbb{E} \|Z\|^2$, but shifted by a factor 2 in $\gamma$, as predicted by section F.3.

H   LARGE VERSION OF THE MICKEY MOUSE I-DIAGRAM

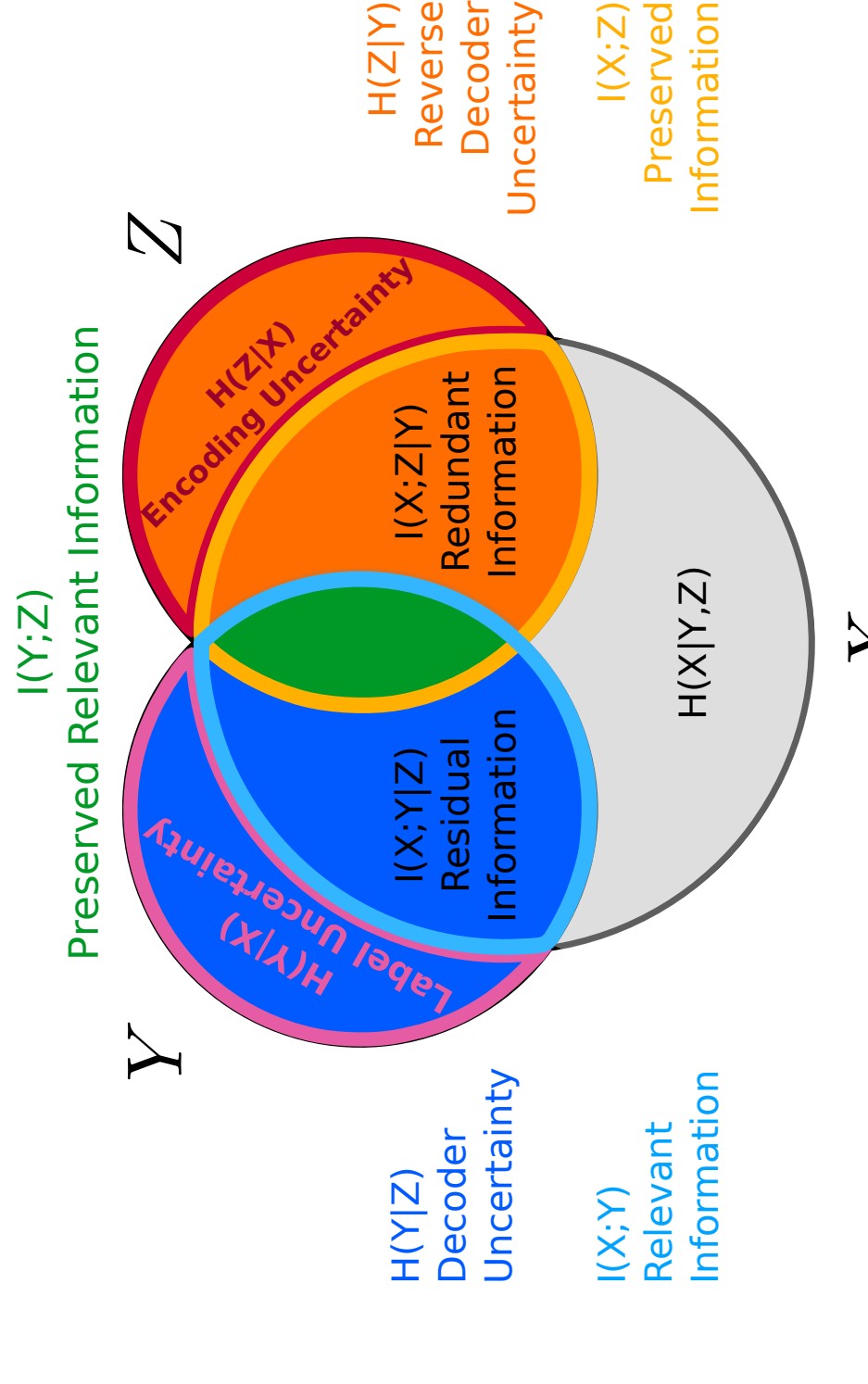

Figure H.1: *Mickey Mouse I-diagram.* See figure 2 for details.

