# OpenReview forum: "Unpacking Information Bottlenecks: Surrogate Objectives for Deep Learning"
_ICLR.cc/2021/Conference — Reject_

### Official Review · AnonReviewer4 · 2020-10-28
**Need to better highlight novel contributions and compare/contrast with VIB**

**Rating:** 6
**Confidence:** 4

**Review:**

Summary of paper:
The authors review the information bottleneck (IB) in the context of deep learning. They discuss the obstacles to applying the IB (and a deterministic variant, the DIB) to modern datasets, review approaches to doing so, and introduce their own scalable approach. Their approach introduces practical surrogate objectives for the information regularizer term, and uses dropout as the source of stochasticity. They take advantage of the scalability of their method to train a ResNet with (D)IB on MNIST, CIFAR-10, and ImageNette and study adversarial robustness and evolution in the information plane.

Pros:
1) The surrogate objectives the authors introduce allow the application of IB without restricting the latent distribution to take on a form with analytic entropy (i.e. gaussian), as is the case the deep variational IB (VIB).
2) The authors are the first, to my knowledge, to scale the DIB from tabular settings (where it was developed) to modern function approximation settings using a clever zero-entropy noise trick (although this did come at the cost of diverging from the deterministic solutions that would be optimal).
3) The authors are the first, to my knowledge, to use dropout as the source of stochasticity that IB requires. This has the advantage of allowing the authors to use nearly arbitrary DNN architectures with (D)IB, as opposed to inserting an explicitly stochastic (gaussian) layer.
4) The paper functions as a good review, independent of the authors’ contributions. A common complaint when reading ML papers is that they don’t discuss related work enough, so this paper was refreshing.

Cons:
1) I struggled to disentangle the novel contributions of the authors from the work they were reviewing. The novel contributions to my knowledge are the first 3 pros above. On the other hand, optimizing decoder uncertainty is not, for example (e.g. it is done by VIB). The authors need to do a better job at highlighting their own contributions but also making clear what is not.
2) I don’t think the authors make a very compelling explicit case for what advantages their approach has over VIB (the main alternative). I do believe there are advantages (see pros 1-3 above), but they are scattered throughout the paper and not always made explicit. I think the authors need a dedicated subsection addressing this. This section should also highlight the disadvantages (looser bound maybe?).
3) Relatedly, why not include direct comparisons to VIB in the experiments? The authors seem to imply that VIB wouldn’t scale to the datasets they tackle, since the experiments in the VIB paper involve pretrained embeddings and smaller models. But a) the VIB paper was written 4 years ago and hardware/software has improved since then b) the model sizes are the same order of magnitude.

Other comments:
1) I thought the heavy use of color distracted more than it helped, though I appreciate the effort.
2) This paper (https://arxiv.org/abs/1712.09657) also attempted to scale DIB to non-tabular problems (although far from the scalability of DNNs). The authors also added noise, but in this case to the data rather than the latents. Different problem being solved, but possibly interesting connection for the authors.

UPDATE

Following the author's response and updated draft, I've raised my score from a 6 to a 7.

UPDATE 2

Following discussion among the reviewers and especially a summary of experimental results by Reviewer 3, I'm lowering score back to a 6.

---

> ### Author Response · Authors · 2020-11-14
> **Response to Reviewer 4**
>
> We would like to thank the reviewer for highlighting the contributions of the paper, in particular the wide applicability of our objectives to a richer class of latent distributions than had been covered by previous work, the scalability and computational efficiency of our approach, and our discussion of related work. We will endeavour to clarify the points listed below to improve the paper in our revision, which we expect to upload by Monday.
>
> ### Specific Replies
>
> > I struggled to disentangle the novel contributions of the authors from the work they were reviewing. The novel contributions to my knowledge are the first 3 pros above. On the other hand, optimizing decoder uncertainty is not, for example (e.g. it is done by VIB). The authors need to do a better job at highlighting their own contributions but also making clear what is not.
>
> We thank the reviewer for highlighting this source of confusion. With regard to the optimization of the Decoder Uncertainty, our principal contribution is to recognize the generality of the reparameterization trick and expand it to Dropout instead of using restrictive parameterized distributions as VIB and CEB do. We further provide additional analysis into the relationship between Prediction and Decoder Cross-Entropy, demonstrating that the two objectives only coincide when a single-sample estimator is used and drawing connections to orthogonal work which obtains similar findings (eg. Rank1 BNNs by Dusenberry et al (2020)).
>
> > I don’t think the authors make a very compelling explicit case for what advantages their approach has over VIB (the main alternative). I do believe there are advantages (see pros 1-3 above), but they are scattered throughout the paper and not always made explicit. I think the authors need a dedicated subsection addressing this. This section should also highlight the disadvantages (looser bound maybe?).
>
> We thank the reviewer for pointing this out and will add a section, comparing the approach to both VIB and CEB in more detail.
>
> > Relatedly, why not include direct comparisons to VIB in the experiments? The authors seem to imply that VIB wouldn’t scale to the datasets they tackle, since the experiments in the VIB paper involve pretrained embeddings and smaller models. But a) the VIB paper was written 4 years ago and hardware/software has improved since then b) the model sizes are the same order of magnitude.
>
> This is an excellent idea, and one that we attempted to implement when running experiments for this submission. Unfortunately, we were unable to replicate the results listed in the VIB paper using the publicly available code for the MNIST dataset https://github.com/alexalemi/vib_demo and so refrained from including these results in our submission. We will include this comparison in the appendix in our revision.
>
> ---
> Dusenberry, Michael W., Ghassen Jerfel, Yeming Wen, Yi-an Ma, Jasper Snoek, Katherine Heller, Balaji Lakshminarayanan, and Dustin Tran. "Efficient and Scalable Bayesian Neural Nets with Rank-1 Factors." arXiv preprint arXiv:2005.07186 (2020).

---

> > ### Comment · AnonReviewer4 · 2020-11-19
> > **Why does every comment need titled, this is so unnatural**
> >
> > Thank you to the authors for their response as well as the updated draft. I'm satisfied enough to raise my score to a 7.

---

### Official Review · AnonReviewer1 · 2020-10-30
**Interesting paper but could use some clarifications**

**Rating:** 6
**Confidence:** 4

**Review:**

Overview: The authors provide a detailed analysis of the information bottleneck principle to explain how neural networks train and generalise. Specifically, since multiple competing IB objectives exist, the authors develop universal surrogate objectives that are easier to optimise and apply these to several neural network architectures for imaging tasks.

Quality and Clarity: The paper is clearly  well written. I particularly like the use of colour to match corresponding terms in each of the objectives which makes it easy to pin-point which pieces correspond.

Significance: The IB principle is a very useful and relevant concept for specifically optimising models to retain only the relevant information wrt a particular context or prediction task. It has been applied in several contexts eg deep generative models for identifying novel molecule structures (Wieser et al 2020) or deducing sufficient adjustment sets for causal inference (Parbhoo et al 2020) as well as DNNs in general (Tisbhy and Zaslavksy, 2015). Since it relies entirely on information theoretic quantities, it is widely applicable across several domains. Analysing these information theoretic objectives in order to make sense of these models is very important.

Pros:  1) The work presents a very rigorous analysis and discussion of multiple competing IB objectives and discusses the implications of each of these.

2) The authors present tractable surrogate objectives that can make optimisation easier. Since these are defined entirely in terms of entropies as well, the work is applicable across various kinds of domains -- a key advantage of the classic IB too.

Cons:

1) I think the authors do a good job in theoretically motivating the particular surrogate objectives, but I would have liked to have seen some discussion as to why using a surrogate objective is sensible in the first place, versus say performing comparisons of deep IBs and VAEs. VAEs use maximum marginal likelihood as an objective. How does using the surrogate objective compare to maximum marginal likelihood? What are the implications of this for downstream application of the IB? Should IBs with surrogate objectives only be used for compression or also for prediction tasks like VAEs?


2) I would have also like to have seen empirically how the surrogate objectives can generalise across domains that are not images? Are these objectives robust in other applications such as EHR data or say the chemical/molecular domain?

3) I also think the generalisation plots are not the most intuitive to understand from first glance and require more parsing and explanation in the text.

---

> ### Author Response · Authors · 2020-11-14
> **Response to Reviewer 1**
>
> We thank the reviewer for highlighting our rigorous analysis and our tractable objectives, easy-to-optimize objectives, which we hope will make IB objectives more accessible to the broader research community. We are also happy that the reviewer appreciates our usage of colors to make it easier to identify the different terms.
>
> We would also like to thank the reviewer for suggesting additional recent applications of IB objectives. We will add these references to the introduction of the paper in our revision, which we expect to upload by Monday.
>
> ### Specific Replies
>
> > I think the authors do a good job in theoretically motivating the particular surrogate objectives, but I would have liked to have seen some discussion as to why using a surrogate objective is sensible in the first place, versus say performing comparisons of deep IBs and VAEs. VAEs use maximum marginal likelihood as an objective. How does using the surrogate objective compare to maximum marginal likelihood? What are the implications of this for downstream application of the IB? Should IBs with surrogate objectives only be used for compression or also for prediction tasks like VAEs?
>
> This is an intriguing point. Indeed, there is a strong connection between IB objectives and VAEs. For example, DVIB and $\beta$-VAEs are related: the $\beta$-VAE objective is essentially DVIB for a generative model (Appendix B of Alemi et al, 2017). More broadly, IB principles have been successfully applied within unsupervised learning as we relate in our introduction (Oord et al., 2018; Belghazi et al., 2018; Zhang et al., 2018; Burgess et al, 2018). We want to consider the effect of our surrogate objective on learnt representations in unsupervised settings in future work..
>
> > I would have also like to have seen empirically how the surrogate objectives can generalise across domains that are not images? Are these objectives robust in other applications such as EHR data or say the chemical/molecular domain?
>
> Evaluating the effect of IB objectives on a broader class of neural network architectures is an avenue we are eager to explore in future work. Our objectives can in principle be easily slotted into any DNN architecture, including architectures optimized for non-visual domains. Further, our objectives can be applied in Bayesian neural networks which use dropout for posterior approximation. We plan to investigate these extensions in future work.
>
> > I also think the generalisation plots are not the most intuitive to understand from first glance and require more parsing and explanation in the text.
>
> We thank the reviewer for highlighting this and will include a clearer explanation in our revision.
>
> ---
> Aaron van den Oord, Yazhe Li, and Oriol Vinyals. Representation learning with contrastive predictive coding. arXiv preprint arXiv:1807.03748, 2018.
>
> Mohamed Ishmael Belghazi, Aristide Baratin, Sai Rajeshwar, Sherjil Ozair, Yoshua Bengio, Aaron Courville, and Devon Hjelm. Mutual information neural estimation. In International Conference on Machine Learning, pages 531–540, 2018.
>
> Ying Zhang, Tao Xiang, Timothy M Hospedales, and Huchuan Lu.   Deep mutual learning. In Proceedings of the IEEE Conference on Computer Vision and Pattern Recognition, pages 4320–4328, 2018.
>
> Burgess, C. P., Higgins, I., Pal, A., Matthey, L., Watters, N., Desjardins, G., & Lerchner, A. (1804). Understanding disentangling in β-VAE. arXiv 2018. arXiv preprint arXiv:1804.03599.

---

> > ### Author Response · Authors · 2020-11-18
> > **An information-theoretic approach to VAEs**
> >
> > Further to the question about the applicability of IB surrogates to VAEs, we can deduce an objective similar to the IB objective for VAEs using the insights from the paper: to obtain an ELBO, we use $H[X] + H[Z|X] = H[X|Z] + H[Z]$ and rearrange:
> > $$ H[X] = H[X|Z] + H[Z] - H[Z|X] \overset{\text{(1)}}{\le} H_\theta[X|Z] + H[Z] - H[Z|X] \overset{(2)}{\le} H_\theta[X|Z] + H[Z].$$
> > We can also put this equation into words: we want to find latent representations such that the reconstruction cross-entropy $H[X|Z]$ and the latent entropy $H[Z]$, which tell us about the length of encoding an input sample, become minimal and approach the true entropy as average optimal encoding length of the dataset distribution.
> >
> > The first inequality (1) stems from introducing a cross-entropy approximation $H_\theta[X|Z]$ for the conditional entropy $H[X|Z]$. The second inequality (2) stems from the injection of zero-entropy noise with a stochastic encoder. For a deterministic encoder, we would have equality. We also note that (1) is the DVIB objective for a VAE with $\beta=1$, and (2) is the DIB objective for a VAE.
> >
> > Finally, we can use one of the surrogates to upper bound $H[Z]$. For optimization purposes, we can substitute the simplified $L_2$ activation regularizer $\mathbb{E} ||Z||^2$ and minimize
> > $$\min_\theta H_\theta[X|Z] + \mathbb{E} ||Z||^2.$$
> > It turns out that this objective is examined amongst others in the recently published Ghosh et al. (2019) as a *CV-VAE*, which uses a deterministic encoder and noise injection with constant variance. The paper derives this objective by noticing that the explicit parameterizations that are commonly used for VAEs are cumbersome, and the actual latent distribution does often not necessarily match the induced distribution (commonly a unit Gaussian) which causes sampling to generate out-of-distribution data. It fits a separate density estimator on $p(z)$ after training for sampling.  The paper goes on to then examine other methods of regularization, but also provides experimental results on CV-VAE, which are in line with VAEs and WAEs. The derivation and motivation in the paper are different and makes no use of information-theoretic principles. Our short principled derivation above shows the power of using the insights from our paper for applications outside of supervised learning, and we are happy that it has been independently validated already.
> >
> > ---
> > Ghosh, Partha, Mehdi SM Sajjadi, Antonio Vergari, Michael Black, and Bernhard Schölkopf. "From variational to deterministic autoencoders." arXiv preprint arXiv:1903.12436 (2019).

---

### Official Review · AnonReviewer3 · 2020-10-30
**bounds on information bottleneck objectives**

**Rating:** 4
**Confidence:** 4

**Review:**

Summary:
This paper makes a theoretical contribution of three "surrogate objectives" for the information bottleneck principle, followed by empirical results on MNIST, CIFAR-10 and ImageNette (a subset of 10 easily classified classes from ImageNet). The objectives assume we add a single simple of zero-entropy noise to each sample of the output z of the stochastic encoder p(z|x), and then give estimators on an upper bound of the information bottleneck objective.

Evaluation:
Overall, this is a fine paper - the introduction is especially well-written and I appreciated the inclusion of all the information plane images of training trajectories. However, the contributions are not sufficiently novel for acceptance at ICLR; this paper is mostly a duplicate of prior work in this area.

I am not convinced that their theoretical contribution is novel - it seems to be a variant (or a specific case) of prior work on Conditional Entropy Bottleneck (CEB) given in Fischer 2020 (https://arxiv.org/pdf/2002.05379.pdf). Their initial insight (Proposition 1) that recognizes that the information bottleneck objective I(XZ) - beta * I(YZ) can be rewritten as H(Y|Z) + beta' * I(XZ|Y), is exactly the insight given in CEB. This paper bounds the I(XZ|Y) term by assuming Z has zero-mean Gaussian noise (which can be chosen such that it is also zero-entropy noise). In contrast, the CEB paper gives a variational bound on the rewritten objective, and when optimizing this bound you sample from the encoder and use the samples to parameterize a distribution (where a gaussian is the simplest choice of distribution). It seems like this paper is producing a special case of CEB for gaussian assumptions on that distribution family.

In addition, the empirical contributions are decidedly not novel. They claim to present "the first successful evaluation of IB objectives on CIFAR-10 and ImageNette", but prior work contains these evaluations: Fischer 2020 (linked above) contains CIFAR-10 results and Fischer and Alemi 2020 (https://arxiv.org/pdf/2002.05380.pdf) contains robustness results on CIFAR-10 and ImageNet, on larger ResNets than the experiments in this paper.

---

> ### Author Response · Authors · 2020-11-14
> **Response to Reviewer 3**
>
> We would like to thank the reviewer for drawing our attention towards the new revision of “The Conditional Entropy Bottleneck” (CEB2020) and the paper “CEB Improves Model Robustness” (CEBR), which we were not aware of. Moreover, we also want to thank the reviewer for their important comment and for appreciating the writing, exposition and visualizations in our paper.
>
> However, we strongly disagree with the claim that our paper is mostly a duplicate of prior work which we detail below in response to the points made in the review.
>
> We will upload a revised version of the paper by Monday to cite this work mentioned above and update the literature review and contributions accordingly. We apologize for claiming to be the first ones to run experiments on CIFAR-10 and higher-dimensional datasets as we were not aware of this recently published work and the new revision of CEB.
>
> ### CEB2020 and CEBR
>
> We thank the reviewer for bringing the papers CEB2020 and CEBR to our attention -- both of these papers along with our paper were submitted to Arxiv within a few weeks of each other, and we failed to catch the update to CEB2020 in our updated literature review. We were previously unaware of CEBR, whose publication in Entropy occurred one week prior to the ICLR submission deadline and so evaded our literature review. We will be happy to cite this work in our revisions, and to update our reference to CEB2020 to address the changes from the 2019 version of the paper.
>
> We have cited the ICLR 2019 submission of CEB and included it in our comparison. It is great work, and we like the insights in regards to the optimal choice of the Lagrange multiplier, which we connect to the Entropy Distance Metric introduced by MacKay (2003) in section C.4 in our appendix. Independently, we had learnt to appreciate I-diagrams as providing principled intuitions, and we were happy to find them in CEB, too. We decided to provide extensive details and explanations in the appendix of UIB to ensure that future readers can learn to appreciate them.
>
> ### Specific Replies
>
> We want to offer corrections to the following claims about our paper and its contributions:
>
> > The objectives assume we add a single simple of zero-entropy noise to each sample of the output z of the stochastic encoder p(z|x), and then give estimators on an upper bound of the information bottleneck objective.
>
> We are not limited to single samples and analyze multi-sample Dropout approaches in Section 3.2 in the paper: we provide an experiment comparing Decoder Cross-Entropy and Prediction Cross-Entropy (which represent the two different Dropout multi-sample approaches) in the appendix in G.3.3 as well as relevant plots in Figure G.11. We will update Section 3.2 to refer to the appendix explicitly.
>
> > I am not convinced that their theoretical contribution is novel - it seems to be a variant (or a specific case) of prior work on Conditional Entropy Bottleneck (CEB) given in Fischer 2020 (https://arxiv.org/pdf/2002.05379.pdf).
>
> Having now reviewed the 2020 version of the CEB paper in more detail as well as “CEB Improves Model Robustness”, **we are confident that our results are distinct from the conditional entropy bottleneck**. Our paper presents a set of lower bounds on the IB objective that are all obtained by a) decomposing the IB objective into its constituent information quantities and b) computing tractable estimators on lower bounds these quantities. The contribution in a) is largely pedagogical and is the starting insight for a number of lower bounds on the IB objective, not limited to CEB. In b) the paper distinguishes itself from CEB in two key respects: first, we introduce a novel set of theoretical results which have intriguing implications independent of their use in formulating our objectives, and second, we present three distinct estimators of lower bounds, allowing us to compute not only a lower bound on the IB objective but also the DIB objective, which VCEB does not aim to approximate. In short, we present cheap and easy-to-implement estimators for a range of IB objectives (not limited to VCEB), thus making it easier for practitioners to incorporate these objectives into pre-existing architectures.
>
> We cite three concrete sources of novelty for our work:
>
> * we provide theoretical motivation for the addition of zero-entropy noise to the latent representation (Proposition 3);
> * we analyze the difference between optimizing $H[Y|X]$ and $H[Y|Z]$ which lead to two different variants of multi-sample Dropout that can be used in conjunction with our optimization objectives; and
> * our objectives, which unify IB and DIB objectives, don’t require variational approximation of the marginal $p(z)$ or conditional $p(z|y)$, making them straightforward to implement on top of existing architectures.

---

> > ### Author Response · Authors · 2020-11-14
> > **Response to Reviewer 3 (Part 2)**
> >
> > > Their initial insight (Proposition 1) that recognizes that the information bottleneck objective I(XZ) - beta * I(YZ) can be rewritten as H(Y|Z) + beta' * I(XZ|Y), is exactly the insight given in CEB. This paper bounds the I(XZ|Y) term by assuming Z has zero-mean Gaussian noise (which can be chosen such that it is also zero-entropy noise). In contrast, the CEB paper gives a variational bound on the rewritten objective, and when optimizing this bound you sample from the encoder and use the samples to parameterize a distribution (where a gaussian is the simplest choice of distribution). It seems like this paper is producing a special case of CEB for Gaussian assumptions on that distribution family.
> >
> > We present **three** distinct objectives in our paper. Of these three objectives, only one is directly comparable to the CEB objective (the objective based on $\log Var[Z|Y]$); the other two are more directly related to information quantities that do not appear in the CEB. Our ‘CEB-like’ objective does indeed overlap with the VCEB objective in the case of a deterministic encoder (in our objective) and the use of an isotropic Gaussian distribution in the VCEB objective, as used in the implementation of CEBR. However, even this subset of our objective is not strictly a special case of VCEB due to point 2 above.
> >
> > Further, the simplified version of VCEB implemented in “CEB Improves Model Robustness” still requires an explicit reverse decoder $b(z|y)$, while UIB does not (point 3 above). Our approach, therefore, also simplifies the method proposed in CEBR, making it more accessible to the broader deep learning community while attaining comparable results to CEBR. Moreover, we found the other objectives ($\log Var[Z]$ and $\mathbb{E}||Z||^2$) to be stabler under optimization.
> >
> > We agree that the insights we use to derive our method bear resemblance to the derivations of a number of IB objectives, including but not limited to VCEB. The paper’s novelty stems from how it translates these insights into a tractable, easy-to-implement objective, and in the theoretical results used to do so. In particular, our paper provides theoretical motivation for the design choices used to obtain the empirical results in CEBR, where isotropic Gaussian noise of fixed variance is added to the latent mean embeddings. This choice is analogous to our architecture (though we also incorporate stochasticity in the latent encoding), and can be motivated as a corollary of our Proposition 3: namely, the use of fixed variance removes the possibility of a class of pathological optimization trajectories.
> >
> > > In addition, the empirical contributions are decidedly not novel. They claim to present "the first successful evaluation of IB objectives on CIFAR-10 and ImageNette", but prior work contains these evaluations: Fischer 2020 (linked above) contains CIFAR-10 results and Fischer and Alemi 2020 (https://arxiv.org/pdf/2002.05380.pdf) contains robustness results on CIFAR-10 and ImageNet, on larger ResNets than the experiments in this paper.
> >
> > The results in “CEB Improves Model Robustness” for CIFAR-10 and ImageNet are impressive, and the CIFAR-10 robustness results are in line with what we report in this paper. We cannot provide ImageNet results as our own computational resources are not sufficient. There are some significant differences between the implementations in these papers, however. Our experiments also include stochastic encoders using Dropout and are easy to train with Adam. They also don’t require additional change to the training schedule.
> >
> > “CEB Improves Model Robustness” does not include stochastic encoders beyond injecting noise for CIFAR-10 and ImageNet and uses special training schedules that anneal the Lagrange multiplier to train the models. For the 2020 version of the CEB paper, we have not been able to determine what kind of encoder the experiments exactly use.
> >
> > We hope we have been able to clarify our contributions and draw attention to the novelty of our work.

---

### Official Review · AnonReviewer2 · 2020-11-04
**A very good paper on Information Bottleneck**

**Rating:** 8
**Confidence:** 3

**Review:**

This paper provides several surrogates for the Information Bottleneck (IB) and Deterministic Information Bottleneck (DIB) loss functions that are more friendly to optimization. For the decoder uncertainty part, the authors show that using Dropout and cross-entropy loss provides an unbiased estimator for the decoder cross-entropy which upperbounds the decoder uncertainty. For the regularization terms in IB/DIB, the authors inject noises to the latent features to lower-bound the conditional entropy of latent representations, and further proposes three types of surrogate objectives for the regularziation terms. Emprical results on CIFAR/ImageNette (a subset of ImageNet of 10 classes) show that the proposed surrogates yield similar behaviours in terms of adversarial robustness and information plane and the scalability of the proposed method.

Strengths of the paper:
- As this paper claims, this is the first work that proposes some surrogate of IB loss functions that can be easily optimized and thus be scaled to large models and datasets (CIFAR/ImageNette). Results on both datasets show similar behavior (adversarial robustness, two-phase information plane) to IB loss based optimization.
- The injection of random noises into the latent representation is interesting and able to enforce lower-bound on the conditional entropy of latent representations, which further induces some surrogates that are optimization-friendly.
- This paper is well-written, fully-prepared and contains a large amount of results that are of wide interests of researchers working in this topic.

I don't have specific criticisms for this paper.

---

> ### Author Response · Authors · 2020-11-14
> **Response to Reviewer 2**
>
> We want to thank the reviewer for finding our paper well-written and for appreciating the breadth of the presented results. We are grateful for their comments and for recognizing our contributions so clearly.
>
> Especially, for recognizing the importance of:
>
> * proposing three simple surrogate objectives that are more friendly to optimization;
> * injecting noise to lower-bound entropies which avoids pathologies; and
> * analyzing the two cross-entropy losses and the connection to multi-sample Dropout.
>
> Our empirical results for CIFAR10 and Imagenette show that we can obtain robustness and IB plane dynamics in line with the IB principle, even though our surrogate objectives are very simple. In particular, our simplest surrogate objective $\mathbb{E} ||Z^2||$ (L2 activation regularization) together with noise injection can be trivially added to existing models. Moreover, it does not depend on $Y$ in any way. Thus, the benefits of injecting random noise in models and regularizing L2 activations will be of interest to practitioners and can be applied beyond supervised methods.

---

### Author Response · Authors · 2020-11-14
**General Response**

We would like to thank all reviewers for their comments, and specifically Reviewer 3 for pointing us to additional related work (“The Conditional Entropy Bottleneck” and the paper “CEB Improves Model Robustness”). We were only aware of an earlier version of “The Conditional Entropy Bottleneck” (which we had included in our literature review already), and will update our discussion of related work to include these additional papers.

Having now reviewed the 2020 version of the CEB paper in more detail as well as “CEB Improves Model Robustness”, we are confident that our results are distinct from prior work. The similarities between our work and CEB go to the same extent as a number of works that provide lower bounds on the IB objective. Our method’s main distinguishing feature is that unlike the general formulations of many other objectives, our approach is specifically designed with *computational efficiency and simplicity* in mind.

Specifically, our objectives differ along 3 principal axes from variational lower bound approaches such as VIB and VCEB:

1. we provide theoretical motivation for the addition of zero-entropy noise to the latent representation (Proposition 3);
2. we analyze the difference between optimizing $H[Y|X]$ and $H[Y|Z]$ which lead to two different variants of multi-sample Dropout that can be used in conjunction with our optimization objectives; and
3. our objectives, which unify IB and DIB objectives, don’t require variational approximation of the marginal $p(z)$ or conditional $p(z|y)$, making them straightforward to implement on top of existing architectures.

Of course, we have updated the paper to reference the two papers and updated the literature review and contributions accordingly. We expect to upload the revision by Monday. We are happy that “CEB Improves Model Robustness” showed results for CIFAR-10 and ImageNet, which makes us optimistic that our results using our simple surrogate objective will also translate from ImageNette to ImageNet, which we could not validate due to computational constraints.

---
Fischer, Ian; Alemi, Alexander A. 2020. "CEB Improves Model Robustness." Entropy 22, no. 10: 1081.
https://arxiv.org/abs/2002.05380

Fischer, Ian. "The conditional entropy bottleneck." arXiv preprint arXiv:2002.05379 (2020).
https://arxiv.org/abs/2002.05379

---

### Decision · Program_Chairs · 2021-01-07
**Final Decision**

**Decision:**

Reject

**Comment:**

We have a very well informed reviewer who strongly feels that this paper is insufficiently novel and significant further discussion on how the paper might be raised to a publishable level with more empirical results.  I will have to side with the more engaged reviewers who feel that the paper should be rejected.

---

> ### Comment · ~Andreas_Kirsch1 · 2021-01-25
> **Quality of the decision notification**
>
> Dear AC,
>
> the quality of your decision notification is very disappointing and makes your decision intransparent.
>
> The very well informed reviewer/"more engaged reviewers" (R3 I suppose) never engaged with our replies during the discussion period.
>
> There is no insight into the discussion with R3 and reasons that made R4 lower their score to a "weak accept", and there has not been significant visible discussion on what "more empirical results" would need to be provided.
>
> You have also not engaged with our comments in any way.
>
> I would assume that it is the role of the AC to summarize the discussion and make the decision appear reasonable. This has not happened.
>
> I hope that the quality of your engagement and decision notification for this paper is a one-off, and you have performed your function better overall. If not, I hope you will be able to improve on this in the future.
>
> It is rather frustrating to spend a lot of time reviewing other papers and writing lengthy reviews to then see an AC behave so.
>
> Best wishes,\
>  Andreas